# *WARP*: On the Benefits of Weight Averaged Rewarded Policies

## Abstract

Reinforcement learning from human feedback (RLHF) aligns large language models by encouraging their generations to have high rewards, using a reward model trained on human preferences. To prevent forgetting of pre-trained knowledge, RLHF usually incorporates a KL regularization; this forces the policy to remain close to its initialization, though it hinders the reward optimization. To address the trade-off between KL and reward, in this paper we introduce a novel alignment strategy named Weight Averaged Rewarded Policies (*WARP*), merging policies in the weight space at three distinct stages. First, it uses the exponential moving average of the policy as a dynamic anchor in the KL regularization. Second, it applies spherical interpolation to merge independently fine-tuned policies into a new enhanced one. Third, it linearly interpolates between this merged model and the initialization, to recover features from pre-training. This procedure is then applied iteratively, with each iteration's final model used as an advanced initialization for the next, progressively refining the KL-reward trade-off, achieving superior rewards at fixed KL. Experiments with Gemma policies validate that *WARP* improves their quality and alignment, outperforming open-source models.

## 1 Introduction

**LLM alignment.** Large language models (LLMs) like Gemini (Gemini Team, 2023) and GPT-4 (OpenAI, 2023), along with their open-weight counterparts (Jiang et al., 2023; Gemma Team et al., 2024), demonstrate remarkable abilities as chatbots, but also for tasks like mathematics and coding (Bubeck et al., 2023). These capabilities largely emerge from pre-training on next-token prediction (Radford et al., 2018; 2019), subsequently refined through supervised fine-tuning (SFT) (Raffel et al., 2020; Wei et al., 2022). As these LLMs become more powerful, aligning them with human values becomes increasingly crucial to ensure safe deployment (Amodei et al., 2016; Hendrycks & Mazeika, 2022). To this end, reinforcement learning from human feedback (RLHF) has become the prominent strategy (Christiano et al., 2017; Ziegler et al., 2019; Stiennon et al., 2020), first learning a reward model (RM) on human preferences, before optimizing the LLM to maximize predicted rewards.

**Challenges in RLHF.** However, RLHF introduces several unresolved challenges (Casper et al., 2023). First, the limited scope of fine-tuning, often restricted to relatively small datasets, can lead to excessive specialization and catastrophic forgetting (French, 1992) of the broad and diverse knowledge acquired during pre-training (Goodfellow et al., 2013; Li & Hoiem, 2017; Kirkpatrick et al., 2017; Kumar et al., 2022). Such *alignment tax* (Ouyang et al., 2022) can degrade the LLM's reasoning capabilities and performance on NLP benchmarks (Dong et al., 2023a; Lin et al., 2024a). Second, maximizing an imperfect RM presents several issues on its own, as the LLM can learn to exploit loopholes in the RM (Clark & Amodei, 2016; Pan et al., 2022) when it deviates significantly from its initialization (Gao et al., 2023). Such *reward hacking* (Askell et al., 2021; Skalse et al., 2022) can produce outputs that are linguistically flawed (Lewis et al., 2017), excessively verbose (Singhal et al., 2023), or sycophantic (Perez et al., 2022; Sharma et al., 2023), thereby raising misalignment (Taylor et al., 2016; Ngo et al., 2022) and safety (Amodei et al., 2016; Hendrycks & Mazeika, 2022) concerns. Finally, RLHF can reduce the diversity of generations (Kirk et al., 2024), potentially leading to policy collapse (Moalla et al., 2024; Hamilton, 2024). Such *loss of diversity* limits use in creative or exploratory tasks and can result in the LLM systematically refusing to answer. Overall, achieving high rewards based on an imperfect RM on a selected distribution of prompts is insufficient due to potential reward misspecification and distribution shifts upon deployment.

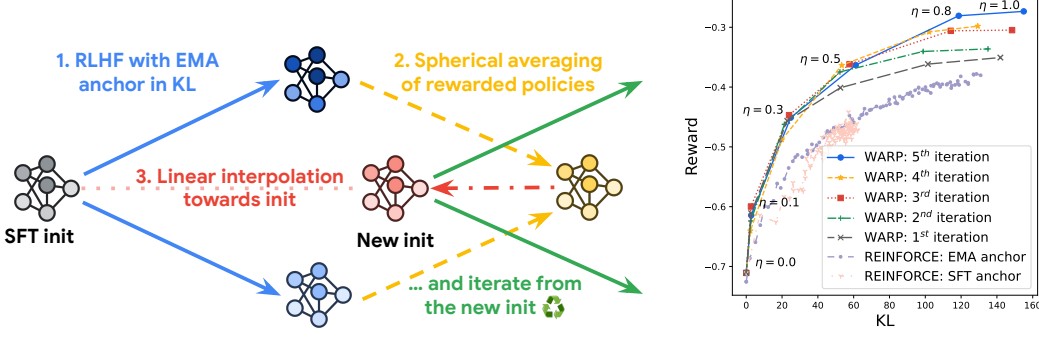

(a) *WARP* with three model merging stages, applicable iteratively.

(b) KL-reward fronts.

Figure 1: Figure 1(a) illustrates the RLHF alignment process with *WARP* from a supervised fine-tuned (SFT) LLM. *WARP* uses model merging by weight averaging at three different stages. First, the exponential moving average (*EMA*) (Izmailov et al., 2018) of the policy serves as the anchor for KL regularization (Jaques et al., 2017). Second, the independently fine-tuned policies are merged by spherical linear interpolation (*SLERP*) (Shoemake, 1985) of their task vectors (Ilharco et al., 2023). Third, we interpolate towards the initialization (*LITI*) (Wortsman et al., 2022b), revealing a Pareto front of solutions as we slide the interpolating coefficient $\eta$ from 1 to 0. This results in the "*WARP*: 1st iteration" curve from Figure 1(b) which improves over the REINFORCE (Williams, 1992) fine-tuning trajectories. Critically, iteratively using a point from this front as an advanced initialization for the next episode *WARP* improves performance. Details in Figure 4(c).

**RL with KL regularization.** To address these issues, previous works constrained the reward optimization by integrating a Kullback-Leibler (KL) regularization (Jaques et al., 2017; Geist et al., 2019), using the SFT initialization as the anchor. As clarified in Section 2, this KL regularization forces the policy to remain close to its initialization (Lazaridou et al., 2020; Lu et al., 2020), mitigating forgetting and reward hacking (Gao et al., 2023). However, employing the SFT model as the anchor may lead to reward underfitting: indeed, there is a fundamental tension between reducing KL and maximizing reward. Thus, different policies should be compared in terms of trade-off between KL-reward as in Figure 1(b), where the $x$-axis is the KL and the $y$-axis is the reward as estimated by the RM, with the optimal policies located in the top-left of the plot.

**On model merging by weight averaging.** To improve the trade-off between KL and reward during RLHF, we leverage the ability to merge LLMs by weight averaging (WA) (Utans, 1996). WA relies on the linear mode connectivity (Frankle et al., 2020; Neyshabur et al., 2020), an empirical observation revealing linear paths of high performance between models fine-tuned from a shared pre-trained initialization. Model merging was shown to improve robustness under distribution shifts (Izmailov et al., 2018; Wortsman et al., 2022a; Ramé et al., 2022) by promoting generalization and reducing memorization (Ramé et al., 2024), to combine models' abilities (Ilharco et al., 2023; 2022; Ramé et al., 2023), to reduce forgetting in continual learning (Stojanovski et al., 2022), to enable collaborative (Raffel, 2023) and distributed (Douillard et al., 2023) learning at scale, without computational overheads at inference time. Model merging is increasingly adopted within the open-source community (Goddard et al., 2024; Lambert & Morrison, 2024), leading to state-of-the-art models in specialized domains (Labrak et al., 2024) but also significant advancements on general-purpose benchmarks (Labonne, 2024b;a). In particular, while WA was initially mostly used for discriminative tasks (Wortsman et al., 2022a) such as reward modeling (Ramé et al., 2024), it is now becoming popular for generative tasks (Rofin et al., 2022; Akiba et al., 2024); its use in KL-constrained RLHF has already shown preliminary successes in a few recent works (Ramé et al., 2023; Noukhovitch et al., 2023; Lin et al., 2024a; Liu et al., 2024; Gorbatovski et al., 2024; Munos et al., 2023), further elaborated in Section 5.

***WARP.*** In this paper, we propose Weight Averaged Rewarded Policies (*WARP*), a simple strategy for aligning LLMs, illustrated in Figure 1(a) and detailed in Section 3. *WARP* is designed to optimize the KL-reward Pareto front of solutions, as demonstrated in Figure 1(b). *WARP* uses three variants of WA at three different stages of the alignment procedure, for three distinct reasons.

**Stage 1: *Exponential Moving Average* (*EMA*).** During RL fine-tuning, instead of regularizing the policy towards the SFT initialization, *WARP* uses the policy's own exponential moving average (Polyak & Juditsky, 1992) as a dynamic updatable anchor in the KL. This stage enables stable exploration with distillation from an EMA teacher (Tarvainen & Valpola, 2017) and annealed constraint.

**Stage 2: *Spherical Linear intERPolation of task vectors* (*SLERP*).** Considering $M$ policies RL fine-tuned independently with their own *EMA* anchor, we merge them by spherical linear interpolation (Shoemake, 1985) of their task vectors (Ilharco et al., 2023). This stage creates a merged model with higher reward by combining the strengths of the $M$ individual policies.

**Stage 3: *Linear Interpolation Towards Initialization* (*LITI*).** Considering the merged policy from *SLERP*, *WARP* linearly interpolates towards the initialization, akin to WiSE-FT (Wortsman et al., 2022b). This stage allows to run through an improved Pareto front simply by adjusting the interpolating coefficient $\eta$ between 1 (high reward but high KL) and 0 (small KL but small reward). Critically, selecting an intermediate value for $0 < \eta < 1$ offers a balanced model that can serve as a new and improved initialization for subsequent iterations of *WARP*.

**Experiments and discussion.** In Section 4, we validate the efficacy of *WARP* for the fine-tuning of Gemma 7B (Gemma Team et al., 2024). Finally, in Section 6, we discuss the connections between *WARP*, the distributed learning literature (Raffel, 2023; Douillard et al., 2023) and iterated amplification (Christiano et al., 2018), illustrating how *WARP* embodies their principles to enable scaling post-training, for continuous alignment and improvement of LLMs.

## 2    CONTEXT AND NOTATIONS

**RL for LLMs.** We consider a transformer (Vaswani et al., 2017) LLM $f(\cdot, \theta)$ parameterized by $\theta$. Following the foundation model paradigm (Bommasani et al., 2021) and the principles of transfer learning (Oquab et al., 2014), those weights are trained via a three-stage procedure: pre-training through next token prediction, supervised fine-tuning resulting in $\theta_{\text{sft}}$, and ultimately, RLHF (Christiano et al., 2017; Ouyang et al., 2022) to optimize a reward $r$ as determined by a RM trained to reflect human preferences. In this RL stage, $\theta$ defines a policy $\pi_\theta(\cdot \mid \boldsymbol{x})$ by auto-regressively generating token sequences $\boldsymbol{y}$ from the prompt $\boldsymbol{x}$. The primary objective is to find weights maximizing the average reward over a dataset of prompts $\mathcal{X}$: $\operatorname{argmax}_\theta \mathbb{E}_{\boldsymbol{x} \in \mathcal{X}} \mathbb{E}_{\boldsymbol{y} \sim \pi_\theta(\cdot|\boldsymbol{x})} \Big[ r(\boldsymbol{x}, \boldsymbol{y}) \Big]$.

**KL vs. reward.** Optimizing solely for $r$ can (i) forget general abilities from pre-training (French, 1992) as an alignment tax (Ouyang et al., 2022; Lin et al., 2024a), (ii) hack the reward (Askell et al., 2021; Skalse et al., 2022) leading to potential misalignment, or (iii) reduce the diversity of possible generations (Kirk et al., 2024) (as visible in Appendix F). To mitigate these risks, a KL regularization is usually integrated to balance fidelity to the initialization and high rewards:

$$\operatorname*{argmax}_{\theta} \mathbb{E}_{\boldsymbol{x} \in \mathcal{X}} \big[ \mathbb{E}_{\boldsymbol{y} \sim \pi_\theta(\cdot|\boldsymbol{x})} r(\boldsymbol{x}, \boldsymbol{y}) - \beta \mathrm{KL}(\pi_\theta(\cdot \mid \boldsymbol{x}) \| \pi_{\theta_{\text{anchor}}}(\cdot \mid \boldsymbol{x})) \big], \tag{1}$$

where usually $\theta_{\text{anchor}} \leftarrow \theta_{\text{sft}}$ and $\beta$ is an hyperparameter, with high values leading to low KL yet also lower reward. The KL-regularized reward function is then $r(\boldsymbol{x}, \boldsymbol{y}) - \beta \log\left(\frac{\pi_\theta(\boldsymbol{y}|\boldsymbol{x})}{\pi_{\theta_{\text{anchor}}}(\boldsymbol{y}|\boldsymbol{x})}\right)$. Our base RL algorithm is a variant of REINFORCE (Williams, 1992). This choice follows recent RLHF works (Roit et al., 2023; Lee et al., 2024a; Ramé et al., 2024) and the findings from Li et al. (2023); Tajwar et al. (2024); Ahmadian et al. (2024) that, in terms of KL-reward trade-off, REINFORCE performs better than the more complex PPO (Schulman et al., 2017) and also better than various offline algorithms such as DPO (Rafailov et al., 2023), IPO (Azar et al., 2023) or RAFT (Dong et al., 2023b). Practitioners then employ early stopping to select an optimal point on the trajectory.

**Weight averaging.** The question of how best to merge models has recently garnered significant attention, driven by the discoveries that deep models can be merged in the weight space (Utans, 1996; Izmailov et al., 2018; Wortsman et al., 2023) instead of in the prediction space, as traditionally done in ensembling (Lakshminarayanan et al., 2017). Specifically, be given two sets of weights $\theta^1$ and $\theta^2$, the different strategies merge them into a new set of weights $\theta$, parameterizing the same non-linear

network architecture. For clarity, we collectively refer to them as weight averaging (WA). The most basic one, uniform linear averaging, is also the most common; in this case, $\theta = \frac{\theta^1 + \theta^2}{2}$.

## 3 WARP

We introduce a novel alignment framework named Weight Averaged Rewarded Policies (*WARP*), illustrated in Figure 1(a) and described in Algorithm 1 below. *WARP* merges LLMs in the weight space to enhance the KL-reward front of policies. The following Sections 3.1 to 3.3 describe the motivations behind applying three distinct variants of WA at the three different stages of *WARP*. In particular, we summarize the key insights as observations, that will be experimentally validated in Section 4 (and in Appendices C and D), and theoretically motivated in Appendix B when possible. Overall, we observe that *WARP* outperforms other RL alignment strategies, without any memory or inference overhead at test time. However, training *WARP* is costly, requiring multiple RL runs at each iteration: see Section 6 for a detailed discussion on the required compute scaling.

---

**Algorithm 1** *WARP* to improve the KL-reward trade-off in alignment

---

**Input:** Weights $\theta_{\text{sft}}$ pre-trained and supervised fine-tuned
   Reward model $r$, prompt dataset $\mathcal{X}$, optimizer Opt
   $I$ iterations with $M$ RL runs each for $T$ training steps
   $\mu$ *EMA* update rate, $\eta$ *LITI* update rate
1: Define $\theta_{\text{init}} \leftarrow \theta_{\text{sft}}$
2: **for** iteration $i$ from 1 to $I$ **do**
3:    **for** run $m$ from 1 to $M$ **do**                                         ▷ Run in parallel
4:       Define $\theta^m, \theta^m_{\text{ema}} \leftarrow \theta_{\text{init}}$
5:       **for** step $t$ from 1 to $T$ **do**
6:          Generate completion $\boldsymbol{y} \sim \pi_{\theta^m}(\cdot \mid \boldsymbol{x})$ for $\boldsymbol{x} \in \mathcal{X}$
7:          Compute $r_\beta(\boldsymbol{y}) \leftarrow r(\boldsymbol{x}, \boldsymbol{y}) - \beta \log \frac{\pi_{\theta^m}(\boldsymbol{y}|\boldsymbol{x})}{\pi_{\theta^m_{\text{ema}}}(\boldsymbol{y}|\boldsymbol{x})}$             ▷ KL regularized reward
8:          Update $\theta^m \leftarrow \text{Opt}(\theta^m, r_\beta(\boldsymbol{y})\nabla_\theta[\log \pi_{\theta^m}(\boldsymbol{y} \mid \boldsymbol{x})])$             ▷ Policy gradient
9:          Update $\theta^m_{\text{ema}} \leftarrow (1 - \mu) \cdot \theta^m_{\text{ema}} + \mu \cdot \theta^m$        ▷ Equation (*EMA*): update anchor
10:      **end for**
11:    **end for**
12:    Define $\theta^i_{\text{slerp}} \leftarrow \text{slerp}\big(\theta_{\text{init}}, \{\theta^m\}_{m=1}^M, \lambda = \frac{1}{M}\big)$     ▷ Equation (*SLERP*): merge $M$ weights
13:    Update $\theta_{\text{init}} \leftarrow (1 - \eta) \cdot \theta_{\text{init}} + \eta \cdot \theta^i_{\text{slerp}}$        ▷ Equation (*LITI*): interpolate towards init
14: **end for**
**Output:** KL-reward front of weights $\big\{(1 - \kappa) \cdot \theta_{\text{sft}} + \kappa \cdot \theta^I_{\text{slerp}} \mid 0 \leq \kappa \leq 1\big\}$

---

### 3.1 STAGE 1: EXPONENTIAL MOVING AVERAGE AS A DYNAMIC ANCHOR IN KL REGULARIZATION

*EMA* **anchor.** RLHF algorithms typically use the SFT initialization as a static anchor (Jaques et al., 2017; Roit et al., 2023) in the KL regularization, but in RL (notably for control tasks) it is common to regularly update the anchor (Schulman et al., 2015; Abdolmaleki et al., 2018). In this spirit, *WARP* uses the policy's own exponential moving average (*EMA*) (Polyak & Juditsky, 1992), updated throughout the RL fine-tuning process such as, at each training step with $\mu = 0.01$:

$$\theta_{\text{ema}} \leftarrow (1 - \mu) \cdot \theta_{\text{ema}} + \mu \cdot \theta_{\text{policy}}. \tag{EMA}$$

Using $\theta_{\text{ema}}$ as the anchor $\theta_{\text{anchor}}$ in Equation (1) provides several benefits, outlined below.

**Observation 1** (*EMA*). *Policies trained with an exponential moving average anchor benefit from automatic annealing of the* KL *regularization and from distillation from a dynamic mean teacher (Tarvainen & Valpola, 2017). Empirical evidence in Section 4.1.*

**Benefits from *EMA*.** Unlike a static SFT anchor, the dynamic nature of an *EMA* anchor induces a gradual automatic annealing and relaxation of the KL regularization. Specifically, the policy is initially strongly tied to the SFT initialization, and then progressively unleashed, allowing for more aggressive gradient updates later in training, leading to higher rewards. Moreover, by progressively

incorporating knowledge from the training, *EMA* acts as slow weight (Stojanovski et al., 2022; Lee et al., 2024b), and thus performing better than the initialization. But, by also maintaining essential information from the initialization, *EMA* can even perform better than the final policy's weights; studies (Szegedy et al., 2016; Izmailov et al., 2018; Arpit et al., 2021) (see Morales-Brotons et al. (2024) for a review), and specifically (Kaddour, 2022) within the context of LLMs, indicate that averaging checkpoints over steps improves internal representations and thus predictions. Then, *EMA* guides the policy by KL distillation (Hinton et al., 2015) of high-quality target predictions, akin to a mean teacher (Tarvainen & Valpola, 2017) for self-supervised (Sohn et al., 2020; He et al., 2020; Oquab et al., 2024; Caron et al., 2021; Grill et al., 2020) learning. This also relates to deep RL techniques where *EMA* stabilizes exploration toward a Nash equilibrium (Awheda & Schwartz, 2013; 2016; Gorbatovski et al., 2024; Munos et al., 2023), and approximates mirror descent (Bubeck et al., 2015; Geist et al., 2019; Tomar et al., 2020).

## 3.2 STAGE 2: SPHERICAL LINEAR INTERPOLATION OF INDEPENDENTLY REWARDED POLICIES

***SLERP.*** While *EMA* helps for a single RL and a fixed compute budget, it faces limitations due to the similarity of the weights collected along a single fine-tuning (Ramé et al., 2022). In this second stage, we merge $M$ weights RL fine-tuned independently (each with their own *EMA* anchor). This follows model soups from Wortsman et al. (2022a) and its variants (Ramé et al., 2022; 2023) showing that WA improves generalization, and that task vectors (Ilharco et al., 2023) (the difference between fine-tuned weights and their initialization) can be arithmetically manipulated by linear interpolation (*LERP*) (Utans, 1996). Yet, this time, we use spherical linear interpolation (*SLERP*) (Shoemake, 1985), illustrated in Figure 2 and defined below for $M = 2$:

$$\text{slerp}\big(\theta_{\text{init}}, \theta^1, \theta^2, \lambda\big) = \theta_{\text{init}} + \frac{\sin[(1-\lambda)\Omega]}{\sin \Omega} \cdot \delta^1 + \frac{\sin[\lambda\Omega]}{\sin \Omega} \cdot \delta^2, \qquad (\textit{SLERP})$$

where $\Omega$ is the angle between the two task vectors $\delta^1 = \theta^1 - \theta_{\text{init}}$ and $\delta^2 = \theta^2 - \theta_{\text{init}}$, and $\lambda$ the interpolation coefficient. Critically *SLERP* is applied layer by layer, each having a different angle. In Appendix B.3 we clarify how *SLERP* can be used iteratively to merge $M > 2$ models. To enforce diversity across weights, we simply vary the order in which text prompts $x$ are given in each run: this was empirically sufficient, though other diversity strategies could help, e.g., varying the hyperparameters or the reward objectives (as explored in Figure 18(c)).

**Benefits from *SLERP* vs. *LERP*.** Merging task vectors, either with *SLERP* or *LERP*, combines their abililities (Ilharco et al., 2023). The difference is that *SLERP* preserves their norms, reaching higher rewards than the base models; this is summarized in Observation 2. In contrast, and as summarized in Observation 3, the more standard *LERP* has less impact on reward, but has the advantage of reducing KL; indeed, as shown in Appendix B, *LERP* tends to pull the merged model towards the initialization, especially as the angle $\Omega$ between task vectors is near-orthogonal (see Observation 3).

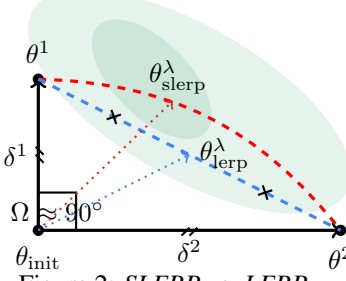

Figure 2: *SLERP* vs. *LERP*.

**Observation 2** (*SLERP*). *Spherical linear interpolation boosts rewards, yet slightly increases* KL. *Empirical evidence in Section 4.2 and theoretical insights in Lemma 1.*

**Observation 3** (*LERP*). *Linear interpolation reduces* KL, *yet has reduced impact on reward. Empirical evidence in Appendix C.1 and theoretical insights in Lemmas 2 and 3.*

**Observation 4** (Task vectors). *Task vectors $\delta$ are close to orthogonal with $\Omega \approx 90°$, while the full weights $\theta$ are collinear. Empirical evidence in Appendix C.2.*

## 3.3 STAGE 3: LINEAR INTERPOLATION TOWARDS INITIALIZATION

***LITI.*** In the previous stage, *SLERP* combines multiple policies into one with higher rewards and slightly higher KL. This third stage, inspired by WiSE-FT from Wortsman et al. (2022b), interpolates from the merged model towards the initialization:

$$\theta^\eta \leftarrow (1 - \eta) \cdot \theta_{\text{init}} + \eta \cdot \theta_{\text{slerp}}. \qquad (\textit{LITI})$$

Adjusting the interpolating coefficient $\eta \in [0, 1]$ trades off between some newly acquired behaviors leading to high rewards vs. general knowledge from the SFT initialization. Specifically, large values

$\eta \approx 1$ provide high rewards but also high KL, while smaller values $\eta \approx 0$ lean towards smaller rewards and minimal KL. Fortunately, we observe that the reduction in KL is proportionally greater than the reduction in reward when decreasing $\eta$. Then, *LITI* empirically yields Pareto fronts that are noticeably above the "diagonal", but also above those revealed during the base RL training runs.

**Observation 5** (*LITI*). *Interpolating weights towards the initialization reveals a better Pareto front than the one revealed during RL fine-tuning. Empirical evidence in Figure 1(b) and Section 4.3, and theoretical insights in Lemmas 4 and 5.*

**Benefits from *LITI*.** Previous works tried to understand how weight interpolation can mitigate forgetting while increasing robustness and generalization. Wortsman et al. (2022b) argue that WiSE-FT, akin to *LITI* in supervised learning contexts, recovers generalizable features from pre-training that might be lost during fine-tuning (Kumar et al., 2022), consistently with WA reducing catastrophic forgetting (Stojanovski et al., 2022; Eeckt et al., 2022) in continual learning. Then in the context of RL, Lin et al. (2024a) argue that *LITI* increases feature diversity, efficiently balancing between generality and task specificity. Finally, Jang et al. (2024) argues that the geometric projection of the ideal weights is located between the merged model and the initialization.

### 3.4 ITERATIVE *WARP*

**Iterative training.** The model merging strategies previously described not only establish an improved Pareto front of solutions, but also set the stage for iterative improvements. Indeed, if the computational budget is sufficient, we can apply those three stages iteratively, using $\theta^\eta$ from previous Pareto front (usually with $\eta = 0.3$, choice ablated in Appendix D.3) as the initialization $\theta_{\text{init}}$ for the next iteration, following the model recycling (Don-Yehiya et al., 2023; Ramé et al., 2023) strategies. Then, the entire training procedure is made of multiple iterations, each consisting of those three stages, where the final weight from a given iteration serves as an improved initialization for the next one.

**Observation 6** (Iterative *WARP*). *The application of* WARP *iteratively progressively refines the Pareto front. Empirical evidence in Sections 4.4 and 4.5.*

## 4 EXPERIMENTS: ON THE BENEFITS OF *WARP*

**Setup.** We consider the Gemma 7B (Gemma Team et al., 2024) LLM, which we seek to fine-tune with RLHF into a better conversational agent. We use REINFORCE (Williams, 1992) policy gradient to optimize the KL-regularized reward. The dataset $\mathcal{X}$ contains conversation prompts. We generate on-policy samples with temperature 0.9, batch size of 128, Adam (Kingma & Ba, 2015) optimizer with learning rate $10^{-6}$ and warmup of 100 steps. *SLERP* is applied independently to the 28 layers. Except when stated otherwise, we train for $T = 9k$ steps, with KL strength $\beta = 0.1$, *EMA* update rate $\mu = 0.01$, merging $M = 2$ policies uniformly $\lambda = 0.5$, and *LITI* update rate $\eta = 0.3$; we analyze those values in Appendix D. We rely on a high capacity reward model, whose architecture is an order of magnitude larger than our policy.

**Summary.** In our experiments, we analyze the KL to the SFT policy (reflecting the forgetting of pre-trained knowledge) and the reward (evaluating alignment to the RM). In Section 4.1, we first show the benefits of using an *EMA* anchor; then in Section 4.2, we show that merging policies trained independently helps. Moreover, in Section 4.3, we show that *LITI* improves the KL-reward Pareto front; critically, repeating those three *WARP* stages can iteratively improve performances in Section 4.4. A limitation is that our RM accurately approximates true human preferences only in low KL region (Gao et al., 2023). Therefore, we finally report other metrics in Section 4.5, specifically comparing against open-source baselines such as Mixtral (Jiang et al., 2024), and reporting performance on standard benchmarks such as MMLU (Hendrycks et al., 2020).

### 4.1 STAGE 1: EXPONENTIAL MOVING AVERAGE AS A DYNAMIC ANCHOR IN KL REGULARIZATION

In Figures 3(a) and 3(b), we compare the training trajectories of different REINFORCE variants, where the changes lie in the choice of the anchor in the KL regularization and of the hyperparameter $\beta$ controlling its strength. Results are computed every 100 training steps. In our proposed version, the anchor is the *EMA* of the trained policy with $\beta = 0.1$ and an *EMA* update rate $\mu = 0.1$ (other

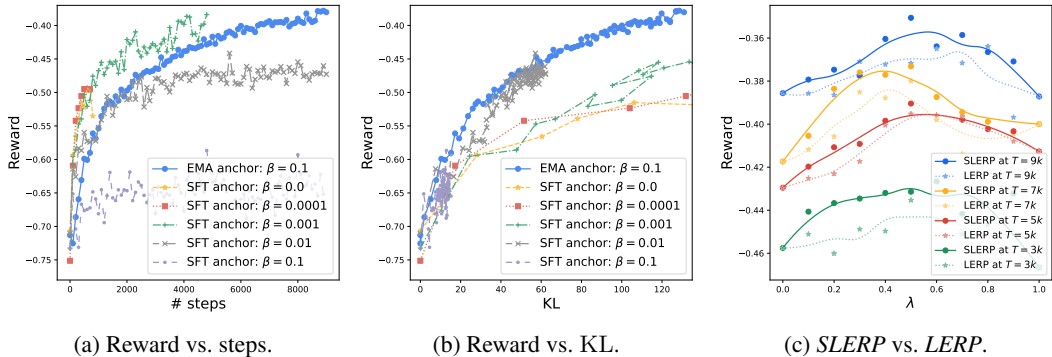

(a) Reward vs. steps.

(b) Reward vs. KL.

(c) *SLERP* vs. *LERP*.

Figure 3: ***EMA* and *SLERP* experiments.** We first compare RL runs with different anchors and strengths $\beta$ in the KL regularization. We show their results along training in Figure 3(a), and their KL-reward Pareto fronts in Figure 3(b). We perform evaluation every 100 steps, and train them for $T = 9k$ steps, though we stopped the trainings if they ever reach a KL of 200 (e.g., after $T = 1k$ training steps when $\beta = 0.0$). Figure 3(c) plots the reward obtained when merging two policies (trained independently after $T$ RL steps with their own *EMA* anchor) with interpolating coefficient $\lambda$; highest rewards are with *SLERP* for $\lambda = 0.5$ and $T = 9k$ steps.

values are ablated in Figure 15). As the Pareto front for our strategy is above and to the left in Figure 3(b), this confirms the superiority of using such an adaptive anchor. The baseline variants all use the SFT as the anchor, with different values of $\beta$. The lack of regularization ($\beta = 0.0$) leads to very fast optimization of the reward in Figure 3(a), but largely through hacking, as visible by the KL exploding in just a few training steps in Figure 3(b). In contrast, higher values such as $\beta = 0.1$ fail to optimize the reward as regularization is too strong, causing a quick reward saturation around $-0.62$ in Figure 3(a). Higher values such as $\beta = 0.01$ can match our *EMA* anchor in low KL regime, but saturates around a reward of $-0.46$. In contrast, as argued in Observation 1, the dynamic *EMA* anchor progressively moves away from the SFT initialization, causing implicit annealing of the regularization. In conclusion, relaxing the anchor with *EMA* updates improves the trade-off between KL and reward, at any given KL level, for a fixed compute budget. We refer the interested reader to additional experiments in Figure 14 from Appendix D.2 where we compare the trained policies with their online *EMA* version.

### 4.2 STAGE 2: SPHERICAL LINEAR INTERPOLATION OF INDEPENDENTLY REWARDED POLICIES

In Figure 3(c), we plot $\lambda \rightarrow r\big(\text{slerp}\big(\theta_{\text{init}}, \theta^1, \theta^2, \lambda\big)\big)$ showing reward convexity when interpolating policies via *SLERP*, validating Observation 2. This mirrors the linear mode connectivity (Frankle et al., 2020) property across weights fine-tuned from a shared initialization, i.e., the fact that interpolated weights perform better than the initial models (recovered for $\lambda = 0$ or $\lambda = 1$). Moreover, *SLERP* consistently obtains higher rewards than *LERP*; yet, this is at slightly higher KL, as further detailed in Appendices B and C.1, where we analyze respectively their theoretical and empirical differences.

### 4.3 STAGE 3: LINEAR INTERPOLATION TOWARDS INITIALIZATION

In Figure 4(a), we merge policies trained for $T$ steps, and then apply the *LITI* procedure. Critically, sliding the interpolating coefficient $\eta \in \{0, 0.1, 0.3, 0.5, 0.8, 1.0\}$ reveals various Pareto fronts, consistently above the training trajectories obtained during the two independent RL fine-tunings. Interestingly, longer fine-tunings improve performances, at high KL, but also at lower KL, simply by using a smaller $\eta$ afterwards. Then in Figure 4(b), we report the Pareto fronts when merging up to $M = 5$ weights. We note that all Pareto fronts revealed when applying *LITI* are consistently above the ones from RL fine-tunings, validating Observation 5. More precisely, best results are achieved by merging an higher number of policies $M$, suggesting a promising scaling direction.

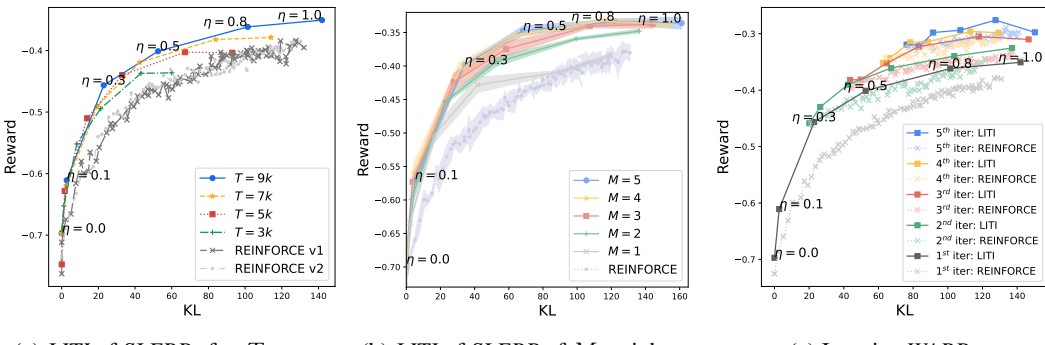

(a) *LITI* of *SLERP* after $T$ steps.      (b) *LITI* of *SLERP* of $M$ weights.      (c) Iterative *WARP*.

Figure 4: ***LITI* and iterative experiments.** Figure 4(a) considers the *LITI* of the *SLERP* of $M = 2$ policies after $T$ steps with $\lambda = 0.5$, interpolating towards their SFT init as we slide $\eta$, revealing Pareto fronts above the $M = 2$ REINFORCE training trajectories. Then Figure 4(b) plots the *LITI* of the *SLERP* of $M$ weights with $\lambda = \frac{1}{M}$ after $T = 9k$ steps: light-colored areas show standard deviations across 5 experiments. The iterative *WARP* procedure is illustrated in Figure 4(c); we fine-tune $M = 2$ policies with their own *EMA* as the anchor, merge them with *SLERP*, interpolate towards their init with *LITI*, and iteratively leverage the weights obtained with $\eta = 0.3$ as the new initialization for the next iteration.

### 4.4 ITERATIVE *WARP*

In Figure 4(c), we apply the iterative procedure described in Section 3.4. At each of the $I = 5$ iterations we train $M = 2$ policies for $T$ steps, with $T = 9k$ for the first iteration, and $T = 7k$ for iterations 2 and 3, and then $T = 5k$ for computational reasons. The *LITI* curves interpolate towards their own initialization (while Figure 1(b) interpolated towards the SFT initialization, see Appendix D.4 for a comparison). We systematically observe that *LITI* curves are above the RL training trajectories used to obtain the inits. Results get better at every iteration, validating Observation 6, although with reduced returns after a few iterations.

### 4.5 COMPARISONS AND BENCHMARKS

**Side by side comparisons.** To conclude our experiments, we compare our trained policies against Mistral (Jiang et al., 2023) and Mixtral (Jiang et al., 2024) LLMs. Each policy generates a candidate answer on an held-out collection of prompts. Then we compute side by side preference rates (Zheng et al., 2023) with "much better", "better" and "slightly better" receiving scores of $\pm 1.5$, $\pm 1$, and $\pm 0.5$ respectively (and ties receiving a score of 0). The reported score is then the average over all prompts. A positive score represents better policies. The results in Table 1 validate the efficiency of *WARP*, as our policies are preferred over Mistral variants, and also outperform the two open-sourced Gemma 7B sharing the same pre-training. However, we note that results stagnate after the 3$^{rd}$ iteration.

Table 1: **Side by side comparisons**.

| Methods | Mistral 7B v1 | Mistral 7B v2 | Mixtral 8x7B |
|---|---|---|---|
| Gemma 7B 1.0 | 0.24 | -0.01 | -0.08 |
| Gemma 7B 1.1 | 0.37 | 0.16 | 0.08 |
| REINFORCE *EMA* anchor | 0.37 | 0.16 | 0.07 |
| *WARP*: 1$^{st}$ iter | 0.42 | 0.23 | 0.13 |
| *WARP*: 2$^{nd}$ iter | **0.45** | 0.25 | 0.16 |
| ***WARP*: 3$^{rd}$ iter** | **0.45** | **0.26** | **0.18** |
| *WARP*: 4$^{th}$ iter | **0.45** | 0.25 | 0.16 |
| *WARP*: 5$^{th}$ iter | **0.45** | 0.24 | 0.17 |

Table 2: **Benchmark results**.

| Methods | MBPP | MMLU | GSM8K | MATH | HumanEval | BBH |
|---|---|---|---|---|---|---|
| Gemma 7B 1.1 | 39.0 | 56.4 | 55.6 | 25.6 | 46.9 | 53.1 |
| *WARP* | **45.4** | **57.6** | **66.8** | **31.0** | **50.0** | **58.8** |

**Benchmarks.** Table 2 compares *WARP* (3rd iter) and Gemma 7B 1.1 (Gemma Team et al., 2024) on popular benchmarks in the zero-shot setup: MBPP (Austin et al., 2021) and HumanEval (Chen et al., 2021) benchmarking coding capabilities, MMLU (Hendrycks et al., 2020) assessing STEM knowledge, the GSM8K (Cobbe et al., 2021) and MATH (Hendrycks et al., 2021) benchmarks targeting reasoning abilities, and the Big Bench Hard (BBH) (Suzgun et al., 2022) benchmark evaluating general capabilities through questions that were deemed difficult for frontier LLMs. *WARP* has particularly strong results on mathematics benchmarks, suggesting higher analytical capabilities.Finally, we refer the reader to Appendix G, where we show how the performances in terms of reward-KL are transposed in terms of SxS and accuracy on real-world benchmarks.

## 5 RELATED WORK

**How to merge models.** The most common model merging strategy is *LERP*, initially used to average checkpoints collected along a single run, uniformly (Szegedy et al., 2016; Izmailov et al., 2018) or with an exponential moving average (*EMA*) (Polyak & Juditsky, 1992). Following the linear mode connectivity (Frankle et al., 2020) observation, the model soups variants (Wortsman et al., 2022a; Ilharco et al., 2023; Ramé et al., 2023) linearly interpolate from different fine-tunings; this relies on the shared pre-training, limiting divergence (Neyshabur et al., 2020) such as models remain in constrained weight regions (Gueta et al., 2023), which also suggests that pre-training mitigates the need to explicitly enforce trust regions in gradient updates (Schulman et al., 2015; 2017). Subsequent works such as TIES merging (Yadav et al., 2023) and DARE (Yu et al., 2023) reduce interferences in multi-task setups with sparse task vectors (Ilharco et al., 2023). In contrast, we use *SLERP*, introduced in Shoemake (1985), increasingly popular in the open-source community (Goddard et al., 2024) but relatively underexplored in the academic literature, with limited studies such as Kim et al. (2024). Some tried to align weights trained from scratch (Entezari et al., 2022; Ainsworth et al., 2022) or with different architectures (Wan et al., 2024); yet, the methods are complex, less robust, and usually require additional training.

**Benefits of model merging.** WA boosts generalization by reducing variance (Wortsman et al., 2022a; Ramé et al., 2022), decreasing memorization (Lin et al., 2024b; Zaman et al., 2023; Ramé et al., 2024) and flattening the loss landscape (Cha et al., 2021). Additionally, merging weights combines their strengths (Ilharco et al., 2023), which helps in multi-task setups (Ilharco et al., 2022; Ramé et al., 2023), to tackle forgetting (Stojanovski et al., 2022; Eeckt et al., 2022; Alexandrov et al., 2024) or to provide better initializations (Don-Yehiya et al., 2023), as explored in Jain et al. (2023); Jang et al. (2024); Huang et al. (2024) for iterative procedures in classification tasks. In particular, we considered using the geometric insights from Eq. 2 in Jang et al. (2024); yet, as our task vectors are nearly orthogonal $\Omega \approx 90°$ (see Appendix C.2), using the update rule $\eta \to \frac{2\cos\Omega}{1+\cos\Omega}$ failed. WA is now also used in RL setups (Nikishin et al., 2018; Gaya et al., 2022; Lawson & Qureshi, 2023); for example, *WARM* (Ramé et al., 2024) merges reward models to boost their efficiency, robustness and reliability. Actually, *WARP* is conceived as a response to *WARM*, demonstrating that model merging can tackle two key RLHF challenges; policy learning in *WARP* and reward design in *WARM*. The most similar works are the following, which also explore how WA can improve policy learning. Noukhovitch et al. (2023) propose an iterative approach with the *EMA* as a new initialization for subsequent iterations. Gorbatovski et al. (2024) and Munos et al. (2023) use *EMA* as the reference, but only for direct preference optimization. Ramé et al. (2023) employ *LERP* to improve alignment in multi-objective RLHF when dealing with different objectives; similarly, Xiao et al. (2023) target multi-task setups with *LERP*. Finally, Lin et al. (2024a) and Fu et al. (2024) use model merging to reduce the alignment tax, although without incorporating *EMA* during training, without merging multiple rewarded policies and not iteratively. Critically, none of these works focus on KL as a measure of forgetting, use *EMA* as the anchor in KL, apply *SLERP* or use *LITI* as the initialization

for subsequent RL iterations. In contrast, *WARP* integrates all those elements, collectively leading to an LLM outperforming Mixtral (Jiang et al., 2024).

## 6 DISCUSSION

**Distributed learning for parallelization and open-source.** *WARP* addresses a crucial challenge: aligning LLMs with human values and societal norms, while preserving the capabilities that emerged from pre-training. To this end, we leverage a (perhaps surprising) ability: policies trained in parallel can combine their strengths within a single policy by weight averaging. Then, the distributed nature of *WARP* makes it flexible and scalable, as it is easily parallelizable by enabling intermittent weight sharing across workers. Actually, iterative *WARP* shares similarities with DiLoCo (Douillard et al., 2023): by analogy, the first stage performs inner optimization on multiple workers independently; the second stage merges gradients from different workers; the third stage performs SGD outer optimization with a learning rate equal to $\eta$. More generally, *WARP* could facilitate open-source (Goddard et al., 2024) collaborative training of policies (Raffel, 2023), optimizing resource and supporting privacy in federated learning (McMahan et al., 2017) scenarios; collaborators could train and share their LLMs, while keeping their data and RMs private. In particular, we show in Appendix E that *WARP* can handle diverse objectives.

**Iterated amplification.** *WARP* improves LLM alignment by leveraging the principles of iterated amplification (Christiano et al., 2018) and progressive collaboration of multiple agents. By analogy, model merging via WA acts as an effective alternative to debate (Irving et al., 2018), with agents communicating within the weight space instead of the token space, ensuring that only essential information is retained (Ramé et al., 2024). Then, *WARP* refines the training signal by combining insights and exploration from diverse models, iteratively achieving higher rewards through self-distillation (Tarvainen & Valpola, 2017), surpassing the capabilities of any single agent. If this is the way forward, then an iterative safety assessment would be required to detect and mitigate potential risks early, ensuring that the development remains aligned with safety standards.

**Scaling alignment.** The *WARP* procedure increases the compute training cost by performing multiple fine-tunings at each iteration. Yet, this should be viewed as "a feature rather than a bug". Specifically, by preventing memorization and forgetting, we see *WARP* as a fine-tuning method that can transform additional compute allocated to alignment into enhanced capabilities and safety. This would allow scaling (the traditionally cheap) post-training alignment, in the same way pre-training has been scaled (Hoffmann et al., 2022). Indeed, historically, pre-training efforts have benefited much more from compute scaling, fine-tuning efforts remaining significantly cheaper. Critically for large-scale deployment, the acquired knowledge is within a single (merged) model, thus without inference or memory overhead, in contrast to "more agents" approaches (Li et al., 2024; Wang et al., 2024). Finally, although *WARP* improves policy optimization, it is important to recognize that *WARP* does not address other critical challenges in RLHF (Casper et al., 2023): to mitigate the safety risks (Amodei et al., 2016; Hendrycks & Mazeika, 2022; Hendrycks, 2023) from misalignment (Taylor et al., 2016; Ngo et al., 2022), *WARP* should be part of a broader responsible AI framework.

## 7 CONCLUSION

We introduce Weight Averaged Rewarded Policies (*WARP*), a novel RLHF strategy to align LLMs with three distinct stages of model merging: exponential moving average as a dynamic anchor during RL, spherical interpolation to combine multiple policies rewarded independently, and interpolation towards the shared initialization. This iterative application of *WARP* improves the KL-reward Pareto front, aligning the LLMs while protecting the knowledge from pre-training, and compares favorably against state-of-the-art baselines. We hope *WARP* could contribute to safe and powerful AI systems by scaling alignment, and spur further exploration of the magic behind model merging.

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

# *WARP*: On the Benefits of Weight Averaged Rewarded Policies

## Supplementary material

This supplementary material is organized as follows:

- Appendix A provides additional illustration of the *WARP* procedure.
- Appendix B details theoretical insights on task vectors, *SLERP*, *LERP* and *LITI*.
- Appendix C details empirical insights on task vectors, *SLERP*, *LERP* and *LITI*.
- Appendix D shows the impact of different design choices in *WARP*.
- Appendix E investigates a potential length bias in *WARP*, and how to fix it.
- Appendix F explores the relationship between KL and diversity in generations.
- Appendix G provides additional SxS and benchmark results.

## A  STRATEGY ILLUSTRATION

In Figure 5, we propose an alternative illustration of *WARP*, where the different stages are more detailed than in Figure 1(a). Then in Figure 6, we also refine our illustration showcasing the similarity and difference between *SLERP* and *LERP*.

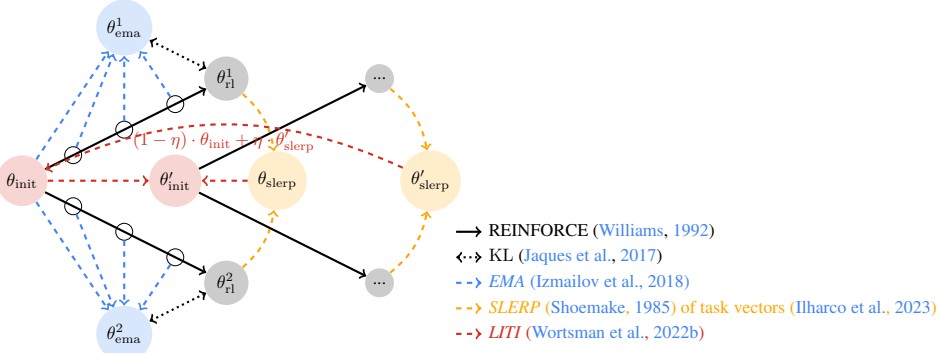

Figure 5: **Detailed illustration of the *WARP* strategy**. From a (pre-trained and supervised fine-tuned) LLM $\theta_{\text{init}}$, we launch $M = 2$ fine-tunings (black arrows ⟶). The innovation of *WARP* lies in the use of model merging by weight averaging at three different stages. First, the exponential moving averages (*EMA*, blue dashed arrows --➤) of the policy (collected at different training steps) serves as the anchor for the KL regularization (black double-headed dotted arrows ⟨⋯➤). The fine-tuned networks are weight averaged using spherical linear interpolation of task vectors (*SLERP*, yellow dashed arrows --➤). Third, we interpolate towards the initialization (*LITI*, red dashed arrows --➤). This obtained model $\theta'_{\text{init}}$ serves as an updated initialization for the next iteration, progressively refining the model's capabilities and alignment. Overall, the final model $\theta'_{\text{slerp}}$ has high reward but also high KL. Then, by interpolation towards the SFT init, we reveal a KL-reward Pareto front of solutions: $\{(1 - \eta) \cdot \theta_{\text{sft}} + \eta \cdot \theta^I_{\text{slerp}} \mid 0 \leq \eta \leq 1\}$.

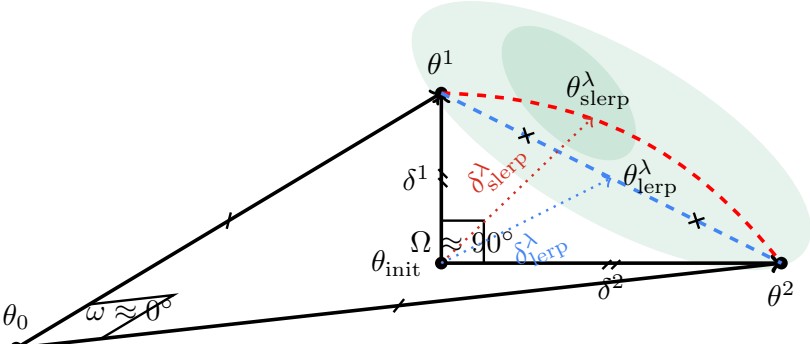

Figure 6: Illustration of the difference between the full weights $\theta^m$ and their task vectors $\delta^m = \theta^m - \theta_{\text{init}}$, where darker areas are of better performance. We found in Appendix C.2 that $\Omega \approx 90°$ where $\Omega$ is the angle between task vectors such as $\cos\Omega = \frac{\delta^1 \cdot \delta_2}{\|\delta^1\|\|\delta^2\|}$, while $\omega$ the angle between the full weights such as $\cos\omega = \frac{\theta^1 \cdot \theta_2}{\|\theta^1\|\|\theta^2\|}$ satisfies $\omega \approx 0°$.

# B THEORETICAL INSIGHTS ON TASK VECTORS, *SLERP*, *LERP* AND *LITI*

Based on the insights from Ilharco et al. (2023) that task vectors (the differences between a fine-tuned model and its initialization) are semantically manipulable and interpretable units in the weight space, we compare *SLERP* and *LERP* merging operations by analyzing their task vectors.

**Background.** Linear interpolation (*LERP*) (Utans, 1996) is the simplest merging strategy, notably used in the model soups variants (Wortsman et al., 2022a), and defined as:

$$\text{lerp}(\theta^1, \theta^2, \lambda) = (1 - \lambda) \cdot \theta^1 + \lambda \cdot \theta^2. \tag{\textit{LERP}}$$

Then, as illustrated in Figure 6, the task vector for *LERP* with interpolating coefficient $\lambda$ is given by: $\delta_{\text{lerp}}^\lambda = \text{lerp}(\theta^1, \theta^2, \lambda) - \theta_{\text{init}} = (1 - \lambda) \cdot \delta^1 + \lambda \cdot \delta^2$. Similarly, we define $\delta_{\text{slerp}}^\lambda = \text{slerp}(\theta_{\text{init}}, \theta^1, \theta^2, \lambda) - \theta_{\text{init}}$ where slerp is defined in Equation (*SLERP*).

## B.1 THEORETICAL INSIGHTS ON THE *SLERP* AND *LERP* TASK VECTORS

We denote $\Omega$ the angle between the the task vectors $\delta^1$ and $\delta^2$:

$$\cos\Omega = \frac{\delta^1 \cdot \delta_2}{\|\delta^1\|\|\delta^2\|}. \tag{2}$$

Based on the empirical observations from Jang et al. (2024), confirmed in our Figure 11(c), we introduce the following Assumption 1 for simplicity.

**Assumption 1** (Task vectors of equal norm). *Independently fine-tuned task vectors have a same norm l:*

$$\|\delta^1\| = \|\delta^2\| = l. \tag{3}$$

**Lemma 1** (*SLERP* task vector). *Under Assumption 1, SLERP preserves the norm of the task vector:*

$$\|\delta_{\text{slerp}}^\lambda\| = l. \tag{4}$$

*Proof.* By definition,

$$\delta_{\text{slerp}}^\lambda = \frac{\sin[(1 - \lambda)\Omega]}{\sin\Omega} \cdot \delta^1 + \frac{\sin[\lambda\Omega]}{\sin\Omega} \cdot \delta^2 \tag{5}$$

Then, as $\delta^1 \cdot \delta^2 = l^2 \cos \Omega$,

$$\frac{\|\delta_{\text{slerp}}^\lambda\|^2}{l^2} = \left(\frac{\sin[(1-\lambda)\Omega]}{\sin \Omega}\right)^2 + 2\frac{\sin[(1-\lambda)\Omega]}{\sin \Omega}\frac{\sin[\lambda\Omega]}{\sin \Omega}\cos(\Omega) + \left(\frac{\sin[\lambda\Omega]}{\sin \Omega}\right)^2 \quad (6)$$

$$= \frac{\sin^2[(1-\lambda)\Omega] + 2\sin[(1-\lambda)\Omega]\sin[\lambda\Omega]\cos(\Omega) + \sin^2[\lambda\Omega]}{\sin^2 \Omega} \quad (7)$$

$$= \frac{\sin^2 \Omega}{\sin^2 \Omega} \quad (8)$$

$$= 1 \quad (9)$$

using trigonometric identities, proving Lemma 1.  □

**Lemma 2** (*LERP* task vector). *Under Assumption 1,* LERP *reduces the norm of the task vector:*

$$\|\delta_{\text{lerp}}^\lambda\| = l\sqrt{1 - 2(1 - \cos \Omega)(\lambda - \lambda^2)}. \quad (10)$$

We recover that averaging weights with $\lambda = 0.5$ tends to reduce the norm of the task vectors, as previously highlighted in Jang et al. (2024).

*Proof.* By definition:

$$\delta_{\text{lerp}}^\lambda = (1 - \lambda) \cdot \delta^1 + \lambda \cdot \delta^2. \quad (11)$$

Then, as $\delta^1 \cdot \delta^2 = l^2 \cos \Omega$,

$$\frac{\|\delta_{\text{slerp}}^\lambda\|^2}{l^2} = (1 - \lambda)^2 + 2\lambda(1 - \lambda)\cos \Omega + \lambda^2 \quad (12)$$

$$= 1 - 2\lambda(1 - \cos \Omega) + 2\lambda^2(1 - \cos \Omega) \quad (13)$$

$$= 1 - 2(1 - \cos \Omega)(\lambda - \lambda^2), \quad (14)$$

proving Lemma 2 when $0 < \lambda < 1$.  □

### B.2 THEORETICAL INSIGHTS ON THE KL

#### B.2.1 LINEAR REGIME

**Assumption 2** (Linear regime (Wortsman et al., 2022b)). *We assume that the predictions of a model $f$, with weights initialized from $\theta_0$ and fine-tuned into $\theta$, can be approximated by first-order Taylor expansion: $\forall \boldsymbol{x}$,*

$$f(\boldsymbol{x}, \theta) \approx f(\boldsymbol{x}, \theta_0) + (\theta - \theta_0) \cdot \nabla_\theta f(\boldsymbol{x}, \theta_0). \quad (15)$$

Assumption 2 defines a neural tangent (Jacot et al., 2018) space in which the relationship between weights and functions is linear. As previously argued in Wortsman et al. (2022a); Ramé et al. (2022), this Taylor expansion is reasonable partly because weights remain close during fine-tunings (Neyshabur et al., 2020), as confirmed in Figure 11 where they have equal norms and a cosine of one. Yet, please note that Ortiz-Jimenez et al. (2023) highlighted some limitations.

#### B.2.2 KL VARIATIONS FOR *LERP*

We consider $\theta^1$ and $\theta^2$ weights fine-tuned from a shared SFT initialization $\theta_{\text{sft}}$. Then in the linear regime from Assumption 2, weight and prediction ensembling behaves similarly:

$$f(\boldsymbol{x}, (1 - \lambda) \cdot \theta^1 + \lambda \cdot \theta^2) \approx (1 - \lambda) \cdot f(\boldsymbol{x}, \theta^1) + \lambda \cdot f(\boldsymbol{x}, \theta^2). \quad (16)$$

This similarity enables to prove the following Lemma 3.

**Lemma 3** (LERP reduces KL). *For an interpolating coefficient $0 \leq \lambda \leq 1$, denoting $\pi_\lambda$ the LERP policy from weight interpolation $(1 - \lambda) \cdot \theta^1 + \lambda \cdot \theta^2$, and $\hat{\pi}_\lambda$ the ensembling policy from prediction interpolation $(1 - \lambda) \cdot \pi_{\theta^1} + \lambda \cdot \pi_{\theta^2}$, then under Assumption 2,*

$$\text{KL}(\pi_\lambda || \pi_{\theta_{\text{sft}}}) \approx \text{KL}(\hat{\pi}_\lambda || \pi_{\theta_{\text{sft}}}) \leq (1 - \lambda)\text{KL}(\pi_{\theta^1} || \pi_{\theta_{\text{sft}}}) + \lambda\text{KL}(\pi_{\theta^2} || \pi_{\theta_{\text{sft}}}), \quad (17)$$

*i.e., the KL for LERP is lower than the interpolated KL.*

*Proof.* The following proof applies the linear assumption and properties of the KL divergence.

**Approximation of KL.** The first approximate equality is a direct application of Assumption 2 to $\pi_\lambda$. Precisely, applying Equation (16) to the definition of $\pi_\lambda = \pi_{(1-\lambda)\theta_1 + \lambda\theta_2}$ yields that $\pi_\lambda \approx \hat{\pi}_\lambda$.

**Upper bound of the KL.** The KL divergence is convex in both its arguments (Csiszár, 1975), thus we directly have that

$$\mathrm{KL}((1-\lambda) \cdot \pi_{\theta^1} + \lambda \cdot \pi_{\theta^2} \| \pi_{\theta_{\mathrm{sft}}}) \le (1-\lambda)\mathrm{KL}(\pi_{\theta^1} \| \pi_{\theta_{\mathrm{sft}}}) + \lambda\mathrm{KL}(\pi_{\theta^2} \| \pi_{\theta_{\mathrm{sft}}}), \qquad (18)$$

which completes the proof.

$\square$

**Remark 1.** *Lemma 3 shows that the LERP $\pi_\lambda$ is closer in* KL *to the original SFT initialization. This relates to Lemma 2, where we show that the linear interpolation reduces the norm to the initialization. As the interpolation brings the weights of the models closer, it is natural that it would also bring the resulting policies closer.*

### B.2.3 KL AND REWARD VARIATION FOR *LITI*

We now consider a given weight $\theta$ (in practice either obtained from *LERP* or *SLERP* of multiple fine-tuned weights) and its associated task vector $\delta = \theta - \theta_{\mathrm{sft}}$. In the linear regime from Assumption 2, for each $\eta \in [0, 1]$, we have the following:

$$f(\boldsymbol{x}, \theta_{\mathrm{sft}} + \eta \cdot \delta) - f(\boldsymbol{x}, \theta_{\mathrm{sft}}) \approx \eta \cdot (f(\boldsymbol{x}, \theta_{\mathrm{sft}} + \delta) - f(\boldsymbol{x}, \theta_{\mathrm{sft}})). \qquad (19)$$

We try to show that:

$$\mathrm{KL}(\pi_{\theta_{\mathrm{sft}}+\eta\cdot\delta} \| \pi_{\theta_{\mathrm{sft}}}) \le \eta \cdot \mathrm{KL}(\pi_{\theta_{\mathrm{sft}}+\delta} \| \pi_{\theta_{\mathrm{sft}}}). \qquad (20)$$

**Lemma 4** (KL upper bound for interpolated distributions)**.** *For an interpolating coefficient $0 \le \eta \le 1$, denoting $\pi_\eta$ the* LITI *policy from weight interpolation $\theta_{\mathrm{sft}} + \eta \cdot \delta$, and $\hat{\pi}_\eta$ the ensembling policy from prediction interpolation $(1 - \eta) \cdot \pi_{\theta_{\mathrm{sft}}} + \eta \cdot \pi_{\theta_{\mathrm{sft}}+\delta}$, then under Assumption 2,*

$$\mathrm{KL}(\pi_\eta \| \pi_{\theta_{\mathrm{sft}}}) \approx \mathrm{KL}(\hat{\pi}_\eta \| \pi_{\theta_{\mathrm{sft}}}) \le \eta\mathrm{KL}(\pi_{\theta_{\mathrm{sft}}+\delta} \| \pi_{\theta_{\mathrm{sft}}}). \qquad (21)$$

*Proof.* The following proof uses the same method as the one of Lemma 3. We use Assumption 2 to link the policy with the interpolation of polices, and the inequality is a result of the KL convexity.

**Approximation of KL.** The first approximate equality is a direct application of Assumption 2 to $\pi_\eta$. Precisely, applying Equation (19) to the definition of $\pi_\eta = \pi_{\theta_{\mathrm{sft}}+\eta\cdot\delta}$ yields that $\pi_\eta \approx \hat{\pi}_\eta$.

**Upper bound of the KL.** Using the fact that the KL is convex, we have

$$\mathrm{KL}(\eta \cdot \pi_{\theta_{\mathrm{sft}}+\delta} + (1 - \eta) \cdot \pi_{\theta_{\mathrm{sft}}} \| \pi_{\theta_{\mathrm{sft}}}) \le \eta\mathrm{KL}(\pi_{\theta_{\mathrm{sft}}+\delta} \| \pi_{\theta_{\mathrm{sft}}}). \qquad (22)$$

$\square$

**Assumption 3** (LITI reward is above the expected reward)**.** *The rewards for the* LITI *interpolated weights are above the interpolated rewards:*

$$r(\pi_0 + \eta \cdot (\pi - \pi_{\theta_{\mathrm{sft}}})) \ge \eta r(\pi) + (1 - \eta)r(\pi_{\theta_{\mathrm{sft}}}), \qquad (23)$$

This Assumption 3 is based on observations from Figure 9(b), and extends to a reward maximization setup the notion of linear mode connectivity (Frankle et al., 2020), usually defined w.r.t. the accuracy in supervised learning.

**Lemma 5** (LITI for KL-reward trade-off)**.** *Be given the convexity of the* KL *from Lemma 4 and the concavity of the reward $r$ in Assumption 3, then the reward vs.* KL *front of* LITI *is above the diagonal. Illustration in Figure 7.*

*Proof.* We obtain a policy $\pi_\theta$ fine-tuned from $\pi_{\theta_{\mathrm{sft}}}$. The *LITI* policy for $\theta_\eta = (1 - \eta) \cdot \theta_{\mathrm{sft}} + \eta \cdot \theta$ is noted $\pi_\eta$. Combining the approximation from Lemma 4 and Assumption 3, we have that

$$r(\pi_\eta) \ge (1 - \eta)r(\pi_{\theta_{\mathrm{sft}}}) + \eta r(\pi_\theta). \qquad (24)$$

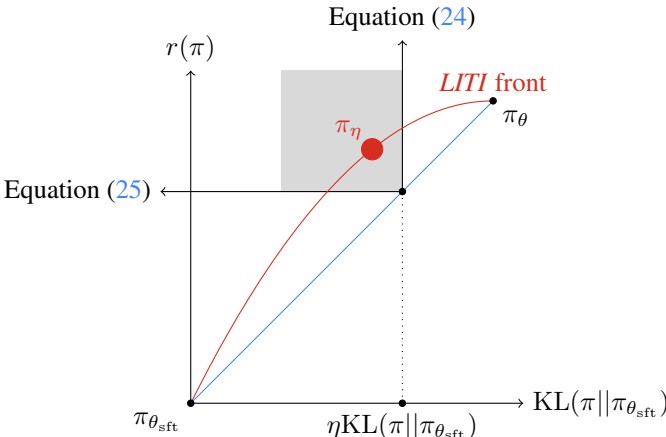

Figure 7: Illustration of Lemma 5. Based on experimental observation and theoretical insights, we see that the Pareto front of the *LITI* policy is better than the identity. It highlights how Equations (24) and (25) place *LITI* policies on the KL-reward plane.

And, from Lemma 4, we also have that

$$\mathrm{KL}(\pi_\eta \| \pi_{\theta_{\mathrm{sft}}}) \leq \eta \mathrm{KL}(\pi_\theta \| \pi_{\theta_{\mathrm{sft}}}). \tag{25}$$

This means that for every *LITI* coefficient $\eta$, the *LITI* policy has a higher reward than the interpolated reward at a lower KL. Geometrically, this means that each point on the Reward-KL front from LITI is on the top left quadrant of the plane according to the corresponding point on the diagonal. □

### B.3 Uniformly averaging $M > 2$ weights with *SLERP*

The *SLERP* merging formula from Equation (*SLERP*) is only defined for $M = 2$ weights. We trivially (and certainly suboptimally) generalize this to $M > 2$ weights in the uniform averaging setup, thus giving an equal coefficient to each of them, i.e., $\lambda = \frac{1}{M}$. In that setup, removing the dependency of $\theta_{\mathrm{init}}$ that is assumed shared, we generalize *SLERP* to merge $M$ weights uniformly through the iterative procedure defined below:

$$\mathrm{slerpm}\big(\{\theta^m\}_{m=1}^M\big) = \mathrm{slerp}\bigg(\mathrm{slerpm}\big(\{\theta^m\}_{m=1}^{M-1}\big), \theta^M, \lambda = \frac{1}{M}\bigg). \tag{26}$$

Though these operations are not associative, the standard deviations in performances are small, as indicated by the shaded areas in Figure 4(b).

## C Empirical insights on task vectors, *SLERP*, *LERP* and *LITI*

### C.1 Empirical insights on the difference between *SLERP* and *LERP*

We now empirically investigate how those theoretical differences between *SLERP* and *LERP* affect the performance of the merged policies.

**SLERP vs. LERP.** In Figure 8 we adjust the interpolating coefficient $\lambda$, highlighting distinct behaviors for *SLERP* and *LERP*. *SLERP* consistently enhances rewards more than *LERP*, as depicted in Figures 3(c) and 8(a). However, a comprehensive evaluation must consider both KL and reward. As shown in Figure 8(b), *LERP* consistently reduces KL, corroborating with Lemma 2 that *LERP* reduces the norm of updates (while *SLERP* preserves it). When plotting these metrics together in Figure 8(c), we observe that *SLERP* and *LERP* target different regions on the Pareto front: *SLERP* achieves higher rewards at the expense of increased KL, while the main impact of *LERP* is to lower KL. This is consistent with Lemmas 2 and 3, be given the orthogonal angles between task vectors $\Omega \approx 90°$ (as shown in Figure 11(a)).

**Combining *SLERP* and *LERP* with *LITI*.** We also compare the behaviours of *SLERP* and *LERP* when we apply *LITI*, as we adjust the interpolating coefficient $\eta$. Figure 9(a) and Figure 9(b) validate that KL is convexe with regard to $\eta$ while the reward is concave with regard to $\eta$, for different values of $M$. This is also highlighted in Figure 10(a), which reproduces the results from Figure 4(b) (and maintaining the same axis limits), replacing *SLERP* by *LERP*: this leads to critical changes in the Pareto fronts. Inded, increasing $M$ now tends to decrease KL for *LERP*, while it used to increase reward with *SLERP*. In Figure 10(b), we explore the extrapolation strategies from (Zheng et al., 2024), using $0 \leq \eta \leq 2$ to compare the full extrapolated fronts from *LERP* and *SLERP*. While both perform similarly on low KL, our results suggest that *SLERP* perform better in high KL regions.

**Conclusion.** *SLERP* demonstrates some key advantages. In particular, it reveals the full Pareto front of solutions, while *LERP* only exposes a portion; extrapolation Figure 10(b) with $\eta > 1$ can partially mitigate this but as our experiments suggest, *LERP* curves consistently lag behind *SLERP* curves in high-reward regions. Moreover, from a practical perspective, *SLERP* scales the choice of $\eta$ effectively, where 1 represents full updates and a fixed value of 0.3 always corresponds to the same operational region, optimizing for high reward and KL.

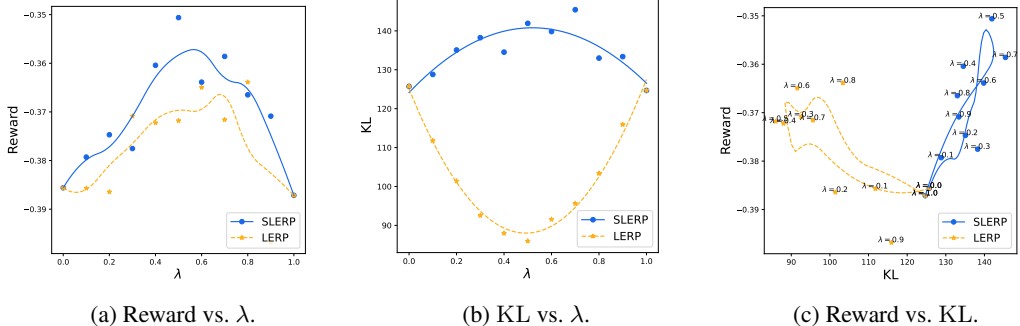

(a) Reward vs. $\lambda$.    (b) KL vs. $\lambda$.    (c) Reward vs. KL.

Figure 8: ***SLERP* vs. *LERP* when sliding the interpolating coefficient** $\lambda$. Considering $M = 2$ weights after $T = 9k$ RL steps, we merge them using either *SLERP* or *LERP*, while sliding the interpolating coefficient $\lambda$ between 0 and 1. We then evaluate the merged checkpoints. Figure 8(a) shows that *SLERP* leads to higher reward than *LERP*, as previously in Figure 3(c). Figure 8(b) shows that *LERP* signicantly reduces the KL (consistently with Lemma 3) while *SLERP* slightly increases it. Figure 8(c) shows how this impact the KL-reward Pareto front, where larger markers/darker colors indicate higher values of $\lambda$; while *SLERP* covers high KL-high reward regions, *LERP* tends to cover regions of lower KL and thus also lower rewards.

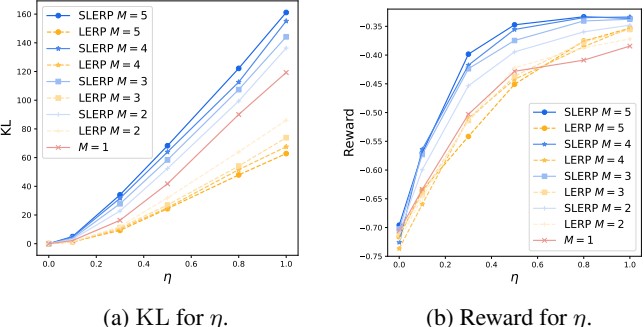

(a) KL for $\eta$.    (b) Reward for $\eta$.

Figure 9: ***SLERP* vs. *LERP* when sliding the interpolating coefficient** $\eta$ **of *LITI*.** In Figure 9(a) we show that the KL is convex (and almost linear) with regard to $\eta$, consistently with Lemma 4. In contrast, Figure 9(b) shows that the reward is concave, validating Assumption 3.

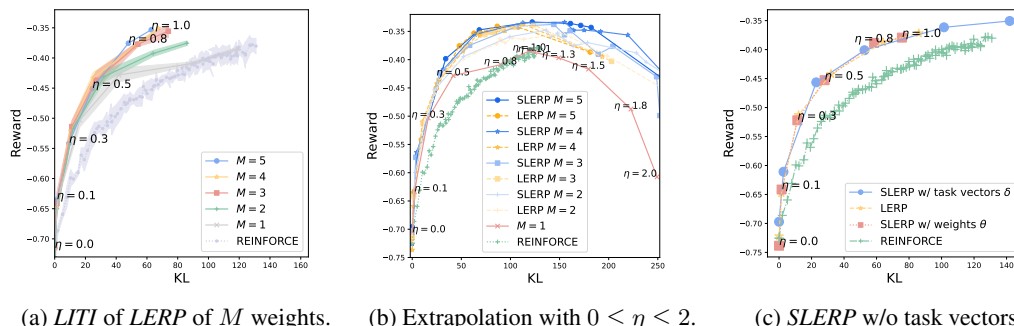

(a) *LITI* of *LERP* of $M$ weights.    (b) Extrapolation with $0 \le \eta \le 2$.    (c) *SLERP* w/o task vectors.

Figure 10: **SLERP vs. LERP when sliding the interpolating coefficient** $\eta$ **of LITI**. Figure 10(a) merges $M$ policies with *LERP* and $\lambda = \frac{1}{M}$ (the endpoints on the top right of the solid lines), and then interpolates towards their SFT init, where light-colored areas show standard deviations across 5 experiments, and with $0 \le \eta \le 1$. In contrast, in Figure 10(b) we investigate extrapolation (Zheng et al., 2024), using $0 \le \eta \le 2$ enabling to compare the full fronts of solutions with *LERP* and *SLERP*. Finally, Figure 10(c) confirms that applying *SLERP* on the full weights $\theta$ rather than on the task vectors $\delta$ perform very similarly to *LERP*.

## C.2 EMPIRICAL INSIGHTS ON THE ROLE OF TASK VECTORS

We now explore the effectiveness of applying *SLERP* on task vectors $\delta$ vs. full weights $\theta$, as illustrated in Figure 6. To this end, in Figure 11 we draw inspiration from Jang et al. (2024) and plot the angles $\Omega$ and $\omega$ and norms of $\delta$ and $\theta$.

**Angles of task vectors** $\Omega \approx 90°$. Figure 11(a) shows that the task vectors are typically orthogonal ($\Omega \approx 90°$), highlighting the diverse trajectories of the different RL fine-tunings. This is in contrast with (Jang et al., 2024) for supervised fine-tunings, where $\Omega$ typically range between $40°$ and $80°$. We suspect that this is related to the underlying differences between reinforcement and supervised learning; in RL the policies are trained on their own generations, creating more orthogonal task vectors, whereas in supervised learning the LLM try to imitate the groundtruth labels, leading to more similar task vectors. The orthogonality of our task vectors prevents the use of the update rule $\eta \to \frac{2\cos\Omega}{1+\cos\Omega}$ suggested from Eq. 2 in Jang et al. (2024), as it would lead to $\eta \approx 0$, deleting any potential update.

**Angles of full weights** $\omega \approx 0°$. In contrast, Figure 11(b) show that full weights remain collinear ($\omega \approx 0°$). This explains the empirical results from Figure 10(c), where applying *SLERP* directly to full weights results in behaviors similar to *LERP*. Indeed, as the angles $\omega \approx 0°$, spherical interpolation effect is minimal because $\sin(x) \approx x + \mathcal{O}(x^3)$, and thus $\frac{\sin[\lambda\omega]}{\sin\omega} \approx \frac{\lambda\omega}{\omega} \approx \lambda$.

**Norms consistency.** Figure 11(c) confirms the consistency in the norms of different task vectors, supporting our Assumption 1. This uniformity is aligned with previous research (Jang et al., 2024). As a side note, this consistency extends to full weights $\theta$, confirming that fine-tuning typically results in minimal changes to the overall weight (Neyshabur et al., 2020).

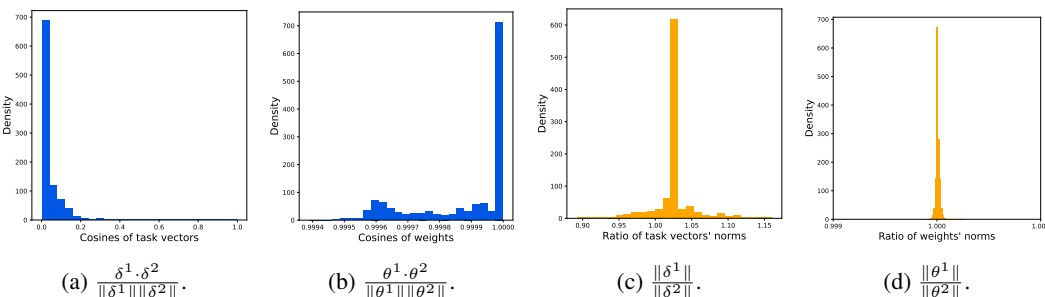

(a) $\frac{\delta^1 \cdot \delta^2}{\|\delta^1\|\|\delta^2\|}$.

(b) $\frac{\theta^1 \cdot \theta^2}{\|\theta^1\|\|\theta^2\|}$.

(c) $\frac{\|\delta^1\|}{\|\delta^2\|}$.

(d) $\frac{\|\theta^1\|}{\|\theta^2\|}$.

Figure 11: **Angles and norms of (full) weights** $\theta^m$ **and their task vectors** $\delta^m = \theta^m - \theta_{\mathrm{init}}$. The histograms are across the 28 layers of the Gemma 7B architecture. Figure 11(a) plots the histograms of task vector cosines. Figure 11(b) plots the histograms of weights cosines. Figure 11(c) plots the histograms of task vector norms ratio. Figure 11(d) plots the histograms of weights norms ratio.

## D   EMPIRICAL INVESTIGATION OF SEVERAL DESIGN CHOICES

We include several experiments showcasing the robustness of *WARP* to different design choices, while further demonstrating its superiority in terms of KL-reward trade-off. Specifically, Appendix D.1 analyzes the performances along training at different steps $T$; Appendix D.2 provides results with different values for the hyperparameters $\mu$ and $\beta$; Appendix D.3 shows the impact of the update rate $\eta$ to provide an improved initialization for the 2nd iteration of *WARP*; finally, Appendix D.4 shows that in iterative *WARP*, interpolating towards the episode initialization or the SFT initialization both perform similarly.

### D.1   ANALYZING THE NUMBER OF TRAINING STEPS

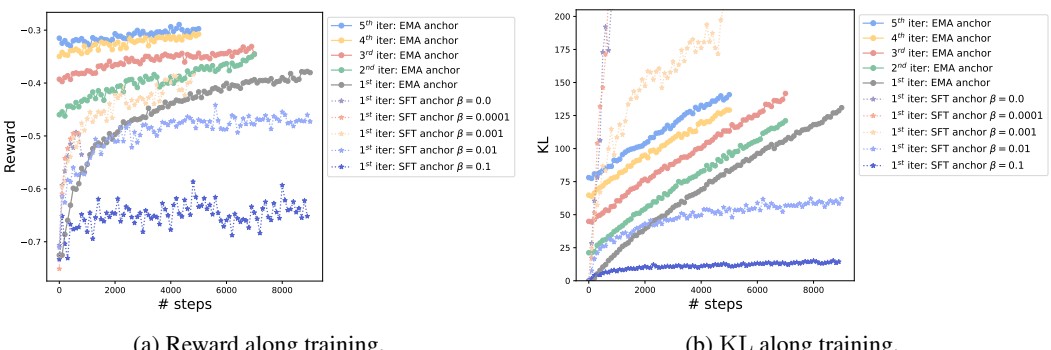

(a) Reward along training.

(b) KL along training.

Figure 12: **Rewards and** KL **at different number of training steps** $T$. Figures 12(a) and 12(b) complement Figure 3(b) and Figure 4(c), this time plotting rewards and KL separately as a function of the number of training steps $T$. Regarding iterative *WARP*, we observe that each iteration has higher rewards but also higher KL (by starting at training step 0 from a new initialization). Regarding the baseline (REINFORCE with SFT anchor), we observe that low values of $\beta$ lead to very fast hacking of the reward, as visible by the KL exploding, while high values of $\beta$ slow down the training.

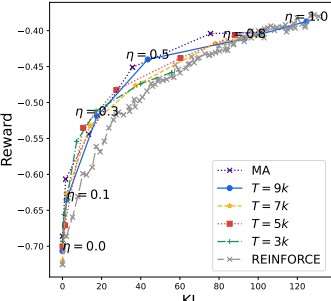

Figure 13: **_LITI_ with $M = 1$ at different number of training steps** $T$. The reward gain is significantly reduced compared to Figure 4(a) where we first merged $M = 2$ policies before applying _LITI_. We also try to perform moving average (MA) (Izmailov et al., 2018) before applying _LITI_, averaging checkpoints collected along a single RL fine-tuning at steps $\{6k, 7k, 8k, 9k\}$; this does not improve performances, suggesting the need to merge weights from independent fine-tunings to have enough diversity (Ramé et al., 2022).

## D.2 ANALYZING THE VALUES OF $\mu$ AND $\beta$

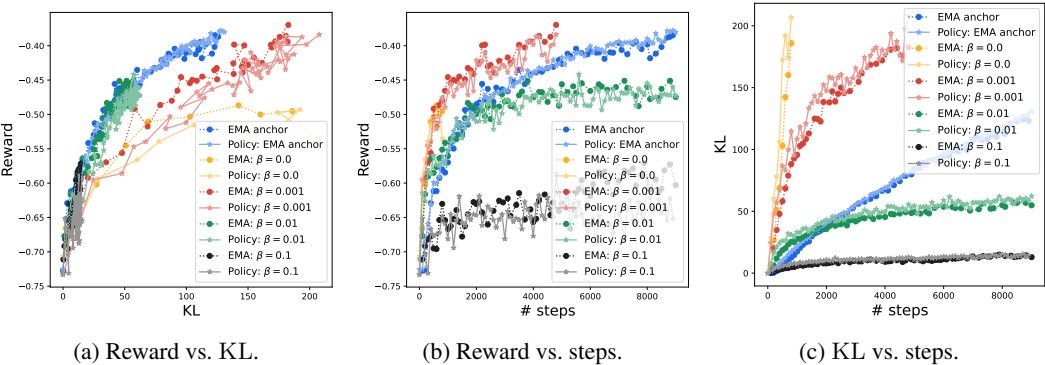

(a) Reward vs. KL.  (b) Reward vs. steps.  (c) KL vs. steps.

Figure 14: **_EMA_ vs. their base policies**, extending Figures 3(a) and 3(b). Figure 14(a) shows that the _EMA_ of all variants (with SFT anchor) perform similarly or better than their base policies in KL-reward. As a reminder, we perform evaluation every 100 steps, and train them for $T = 9k$ steps, though we stopped the trainings if the base policy ever reaches a KL of 200. This confirms Observation 1; the benefits of our variant with _EMA_ anchor is partly explained by distillation from an improved mean teacher (Tarvainen & Valpola, 2017).

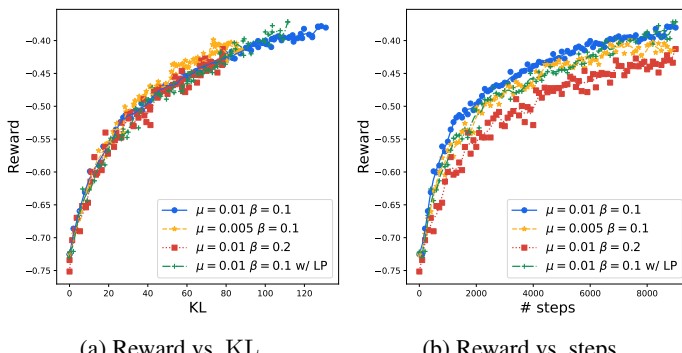

(a) Reward vs. KL.                  (b) Reward vs. steps.

Figure 15: **Experiments ablating the values for the *EMA* update rate** $\mu$ **and the** KL **regularization strength** $\beta$. So far we have systematically used $\mu = 0.01$ and $\beta = 0.1$ for all *EMA*-based runs, including in the iterative *WARP*. These hyperparameters were chosen at the project's onset and have remained unchanged. In Figures 15(a) and 15(b) we increase regularization with $\mu = 0.005$ and $\beta = 0.2$. Our results indicate that reducing $\mu$ or increasing $\beta$ behaves similarly, marginally improving the KL-reward Pareto front but slowing down training. Additionally, we include the training trajectory when using a length penalty (LP), as detailed in Appendix E.

### D.3 ANALYZING THE VALUES OF $\eta$

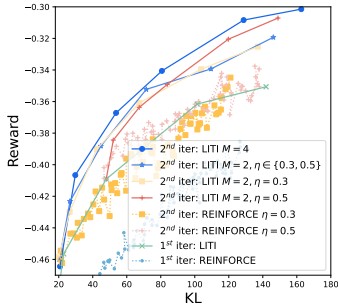

Figure 16: **Experiments ablating the *LITI* update rate** $\eta$. As we initiate the 2nd iteration of *WARP*, selecting an appropriate value for $\eta$ is key, as it determines the starting point $\theta^\eta$ and functions similarly to an outer learning rate (see Section 6). We usually set $\eta = 0.3$, but now provide results with an increased $\eta = 0.5$, starting the 2nd iteration from a more "advanced" position on the previous Pareto front. We run and average $M = 2$ fine-tunings from each of those two initializations for $T = 7k$ steps, before applying *LITI*. Our results indicate that a higher $\eta$ (0.5) performs better in regions of high KL, whereas a lower $\eta$ (0.3) helps in regions with KL below 65. This suggests that the optimal choice for $\eta$ is compute-dependent; a lower $\eta$ is appropriate if further iterations can explore high KL regions, whereas a limited compute budget might benefit from a higher $\eta$. This resembles the learning rate trade-off in optimization, where lower rates improve results but require more training steps. As a final note, we can also use different $\eta$ for the different fine-tunings; notably, we observe that merging all those $M = 4$ RLs perform better (though it doubles the compute).

### D.4 INTERPOLATE TOWARDS THE INITIALIZATION? OR TOWARDS THE SFT?

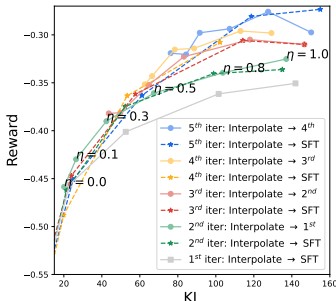

Figure 17: **Experiments ablating the initialization in *LITI*.** We compare LITI either towards the episode-specific initialization (the $\theta^{\eta}$ selected from previous iteration) or towards the SFT (the initialization of the 1st episode). The two resulting fronts are similar. However, in our iterative experiments we interpolate towards the episode-specific initialization as it allows maintaining a constant $\eta$ at each *WARP* iteration, enabling a smooth progression towards the high KL regions.

## E    ADDRESSING LENGTH BIAS IN *WARP*

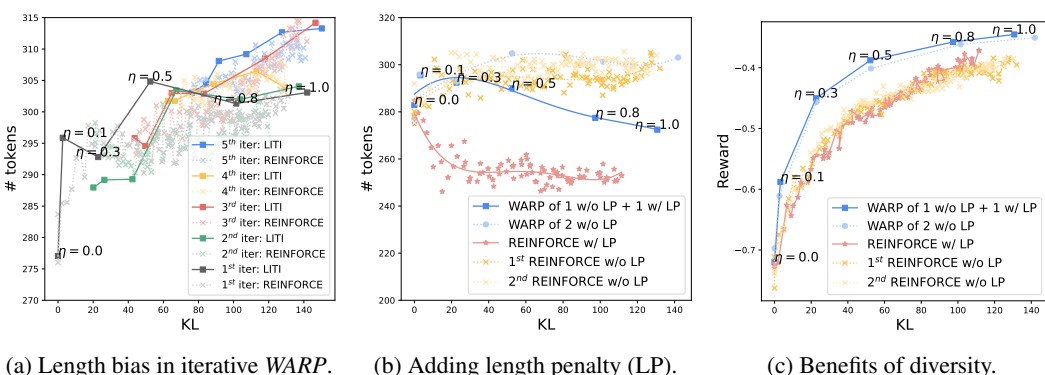

(a) Length bias in iterative *WARP*.    (b) Adding length penalty (LP).    (c) Benefits of diversity.

Figure 18: **Addressing length bias in *WARP*.** Figure 18(a) explores how length and KL change in successive *WARP* iterations. Figure 18(b) demonstrates the effectiveness of length penalty (LP) in reducing output length, and how such policies can merge with others trained without LP. Finally, Figure 18(c) shows that merging policies trained with different objectives further improves the KL-reward trade-off.

**Problem: length bias.** We investigate a potential length bias in *WARP*. LLMs after RLHF tend to be unnecessarily verbose (Shen et al., 2023) because RMs often prefer longer generations to shorter ones, leading to this form of reward hacking. We confirm such a phenomenon in Figure 18(a), where the length of the generation increases with higher KL values. This trend is even more pronounced in iterative *WARP*, where the 3rd iteration generates longer sentences than the 1st iteration at same KL.

**Mitigation strategy: length penalty.** To mitigate this length bias, we integrate a length penalty (LP) into the reward: $-0.0005 \times \text{len}(y)$, following Singhal et al. (2023). From SFT, we launch one RL fine-tuning run with LP, highlighted with red stars in Figure 18(b). This LP leads to shorter outputs as KL increases along training, in contrast to policies trained without LP.

***SLERP* with different configurations.** Figure 18(b) displays the generation lengths from a *SLERP* of two policies, one trained with the LP and the other without. Critically, merging policies from diverse training configurations not only mitigates the length bias but also improves the Pareto front, as illustrated in Figure 18(c). This improvement is likely due to the increased diversity across policies, which appears beneficial for generalization, as shown in supervised learning (Ramé et al., 2022).

**Conclusion.** Those experiments highlight the possibility to fix the length bias, and also the benefits of merging policies trained with diverse rewards.

## F  DIVERSITY IN PREDICTIONS

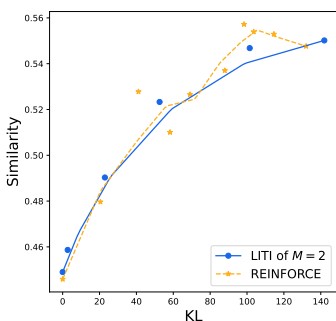

Figure 19: **Confirming diversity loss in RLHF**. The $x$-axis is the KL compared to the SFT initialization; the $y$-axis is the similarity across two generations from a given policy when decoding with temperature $0.9$.

Finally, we investigate the loss in diversity across generations when aligning LLMs, as reported in Kirk et al. (2024). This could have negative consequences for creative or exploratory tasks, or even lead to policy collapse (Moalla et al., 2024; Hamilton, 2024). In Figure 19 we plot the BLEURT similarity (Sellam et al., 2020) across generations, during REINFORCE, or in *LITI* (as we interpolate back towards the SFT initialization). We observe that KL is strongly positively correlated with similarity across generations, confirming that RLHF induces a loss of diversity across generations. This experiment confirms that protecting the KL enables to trade-off between alignment and other benefits from pre-training, such as diversity in generations.

## G  SxS AND BENCHMARK SCORES AT DIFFERENT KL

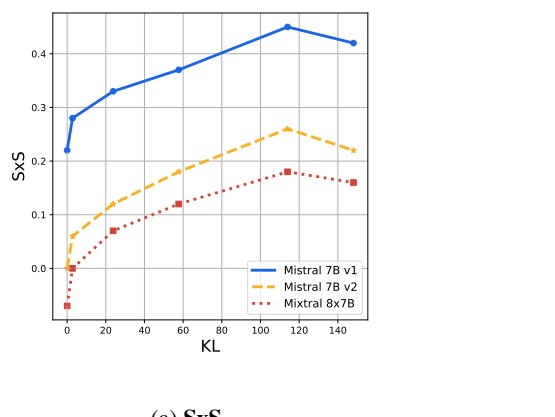

(a) **SxS**.

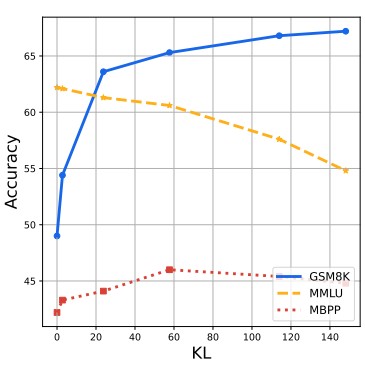

(b) **Benchmark**.

Figure 20: **SxS and benchmark scores at different** KL. The different checkpoints were obtained by LITI between the SFT and the SLERP at the end of the $3^{\text{rd}}$ iter, with coefficients $\eta \in \{0, 0.1, 0.3, 0.5, 0.8, 1.0\}$. In particular, the one with $\eta = 0.8$ was highlighted in Tables 1 and 2. In terms of SxS, hacking appears around around 110 of KL. In terms of accuracies, the alignment tax is benchmark dependent; while GSM8K seems to benefit from RLHF, scores on MMLU significantly reduces while they are stable on MBPP.

