# OpenReview forum: "WARP: On the Benefits of Weight Averaged Rewarded Policies"
_ICLR.cc/2025/Conference — Submitted to ICLR 2025_

### Official Review · Reviewer_HK4g · 2024-10-30

**Soundness:** 3
**Presentation:** 3
**Contribution:** 3
**Rating:** 6
**Confidence:** 3

**Summary:**

The paper proposes a novel alignment technique based on weight average (WA)  for reinforcement learning from human feedback (RLHF) applied to large language models (LLMs). The proposed WARP approach is designed to enhance the balance between reward optimization and the retention of pre-trained knowledge, tackling issues such as catastrophic forgetting and reward hacking often faced in RLHF. The WARP technique integrates three model-merging stages: using the Exponential Moving Average (EMA) as a dynamic anchor in KL regularization, Spherical Linear Interpolation (SLERP) for merging fine-tuned policies, and Linear Interpolation Towards Initialization (LITI) to regain pre-trained features. These steps iteratively refine the KL-reward Pareto front, progressively enhancing model performance. Experimental results on the Gemma 7B model indicate that WARP attains a better Pareto front and achieves superior alignment and generalization across benchmarks, outperforming existing methods.

**Strengths:**

1. **Innovative WA-based Approach to Alignment**: WARP introduces a layered model-merging strategy that addresses KL-reward trade-offs innovatively, leveraging EMA and SLERP in a structured way to boost model alignment with human preferences. Furthermore, the iterative application of WARP stages can progressively refine the KL-Reward Pareto front.
2. **Empirical Validation**: The authors conducted comprehensive experiments to verify the effectiveness of the proposed four algorithmic components: EMA as a dynamic anchor, SLERP, LITI and iterative WRAP. Besides, comparison experiments on the Gemma 7B model show that WARP consistently outperforms open-source models, particularly in alignment and benchmark performance, providing strong evidence for its efficacy.
3. **Scalability and Adaptability**: The WARP process is compatible with distributed learning setups, enabling scalability through parallelizable fine-tuning iterations and potentially making it viable for broader applications in open-source and collaborative environments.

**Weaknesses:**

1. **Increased Computational Cost**: The biggest weakness of WRAP is the computation aspect. The iterative nature of WARP requires multiple RL runs and merging stages, which could be computationally demanding, potentially limiting its practicality for resource-constrained applications.

**Questions:**

1. Can the authors provide insights and more detailed explanations on why interpolating weights towards the initialization can attain a better Pareto front than the one revealed during RLHF?

---

> ### Author Response · Authors · 2024-11-16
>
> We would like to thank R.HK4g for their detailed review, highlighting our contributions and their effectiveness.
>
> ## 1. Increased training cost (duplicate of R.ApSK.3)
>
> We agree that WARP is indeed more costly than traditional single-run strategies; this is the downside of its "scalability and adaptability" to "open-source and collaborative environments", that R.HK4g mentioned in their review.
>
> This limitation was discussed from l.513 to l.520 in the Section 6 of our paper: "we see WARP as a fine-tuning method that can transform additional compute allocated to alignment into enhanced capabilities and safety. This would allow scaling (the traditionally cheap) post-training alignment, in the same way pre-training has been scaled. Indeed, historically, pre-training efforts have benefited much more from compute scaling, fine-tuning efforts remaining significantly cheaper."
>
> For example, the pre-training of the Gemma 7B required 6T tokens, several orders of magnitude above a traditional RLHF run, which would anyway fail to leverage additional compute; if we simply add more training steps, then the KL increases and polices hack the reward and/or suffer from an alignment tax. **WARP is a novel strategy that could enable scaling the post-training phase**. And finally, as stated in conclusion l.535, "we hope WARP could contribute to safe and powerful AI systems by scaling alignment".
>
> ## 2. Benefits of LITI (duplicate of R.ApSK.4)
>
> To explain the success of LITI, we highlighted 3 benefits in Section 3.3 based on previous works:
> 1. following [Wortsman2022b], LITI "recovers generalizable features from pre-training that might be lost during fine-tuning" (l.278).
> 2. following [Lin2024], LITI "increases feature diversity, efficiently balancing between generality and task specificity" (l.279).
> 3. following [Jang2024], LITI approximates the "geometric projection of the ideal weights [which] is located between the merged model and the initialization" (l.281).
>
> Overall, the literature suggests that **LITI efficiently trades-off between the initial abilities of the initialization, and those acquired during fine-tuning**.
>
> We provided novel theoretical insights about it in Lemma 5 from Appendix B.2.3, based on the concavity of the reward and convexity of the KL. As highlighted in Fig.7, we show the "LITI policy has a higher reward than the interpolated reward at a lower KL." (l.1210).
>
> Yet, full theoretical comparison against the checkpoints collected along RLHF would require new theoretical tools. Intuitively, from an optimization perspective, fine-tuning is an unstable procedure with variance and noise; in contrast, the task vector (the difference between the final checkpoint and the init) provides a more consistent direction as it is averaged over multiple batches sampled sequentially. Then, whereas the checkpoints along RLHF suffer from suboptimal trajectory, the LITI depends only on the latest checkpoint, with reduced variance. These findings are consistent with the generalization literature in computer vision, notably Fig.3 from [Wortsman2022b], where LITI is systematically above the optimization trajectory in terms of in-distribution/out-of-distribution trade-off.
>
> [Wortsman2022b] Robust Fine-Tuning of Zero-Shot Models
>
> [Lin2024] Mitigating the Alignment Tax of RLHF
>
> [Jang2024] Model Stock: All We Need Is Just a Few Fine-Tuned Models

---

> > ### Comment · Reviewer_HK4g · 2024-11-27
> >
> > Thank you for the detailed response. The authors' response effectively addressed my question and I decided to keep my positive ratings.

---

### Official Review · Reviewer_pmz9 · 2024-11-04

**Soundness:** 3
**Presentation:** 3
**Contribution:** 3
**Rating:** 6
**Confidence:** 3

**Summary:**

This work gives a novel alignment strategy WARP. This method first uses EMA anchor to train multiple policies with RLHF, then use spherical averaging to average policies, and finally use Linear interpolation to combine with the init model. This process can be repeated. Many experiments have been done to show the effectiveness of this method.

**Strengths:**

1. This work has a good presentation and is easy to understand.

2. The method is shown clearly. The intuition is given and the ablation study for each stage is given.

3. Many experimental results are given to support the method.

**Weaknesses:**

1. I think it is better to add some more introduction for weight averaging. Since I am not quite familiar with this, I may wonder if this is a method  conducting averaging by each parameter or some other techniques.

2. I hope to get some intuition for why WARP can reach a Pareto front, which seems non-trivial for me.

**Questions:**

See weakness part above.

---

> ### Author Response · Authors · 2024-11-16
>
> We would like to thank R.pmz9 for their positive comments on our paper's clarity and our ablation study.
>
> ## 1. Clarity about weight averaging
>
> In the revised version of the paper, you will find **an additional paragraph at the end of Section 2 named "Weight averaging"** clarifying the basics of model merging by weight averaging, which indeed combines all the parameters. This paragraph was previously in the related work Section 5, but moving it earlier in the paper may indeed add more clarity.
>
> Though, we want to point out that we already mentioned this quite early in the paper, notably in the title ("weight averaging"), abstract ("merging policies in the weight space" l.17), introduction ("ability to merge LLMs by weight averaging" l.88), and that Eq.EMA (l.202), Eq.SLERP (l.236) and Eq.LITI (l.265) already showed we merge all the parameters $\theta$.
>
> ## 2. Intuitions for why WARP can improve the Pareto front (duplicate of R.XQNe.2)
>
> As explained in Section 3, WARP can improve the Pareto front thanks to the 3 weight averaging stages, applied iteratively. **For each stage, we provide intuitions, experiments and some theoretical insights**. Specifically,
> - In Section 3.1, paragraph "benefits from EMA", we provide the intuition that having an EMA anchor "allows for more aggressive gradient updates later in training, leading to higher rewards. Moreover, by progressively incorporating knowledge from the training, EMA [...] performs better than the initialization." (l.213-215).
> - In Section 3.2, paragraph "benefits from SLERP vs. LERP.", we provide the intuition that "merging task vectors, either with SLERP or LERP, combines their abilities" (l.246) thus explaining the higher rewards.
> - In Section 3.3, paragraph "benefits from LITI", we provide the intuition that LITI "recovers generalizable features from pre-training that might be lost during fine-tuning" (l.280) and, among other benefits, "increases feature diversity, efficiently balancing between generality and task specificity" (l.282).
> - Finally, in Section 6, paragraph "iterated amplification", we provide the intuition that "WARP improves LLM alignment by leveraging the principles of iterated amplification and progressive collaboration of multiple agents. By analogy, model merging via WA acts as an effective alternative to debate" (l.504-506).
>
> Yet, we also understand R.pmz9's concern about the term "Pareto"; to be clear, we are not claiming that WARP reaches an hypothetical absolute Pareto optimal front of solutions, as it is unknown in real-world applications; we are only claiming that WARP improves the Pareto front previously achievable by other strategies. To prevent any confusion, **we have removed all instances of the terms "Pareto optimal" in the revised version** of the paper; for example in Observation 6, where we replaced "converging to an optimal Pareto front" by "progressively refines the Pareto front" (l.295).

---

### Official Review · Reviewer_XQNe · 2024-11-04

**Soundness:** 3
**Presentation:** 3
**Contribution:** 2
**Rating:** 5
**Confidence:** 3

**Summary:**

This paper studies fine-tuning language models to align with human preferences. It addresses the trade-off between KL divergence and reward, which may help mitigate the alignment tax issue. Specifically, it introduces Weighted Averaged Reward Policies (WARP), merging policies in the weight space using techniques such as exponential moving average and spherical linear interpolation. Experiments on Gemma models demonstrate the effectiveness of the proposed method.

**Strengths:**

- This paper is well-written and easy to follow. Extensive related work is provided to help readers understand the research problem.
- The proposed methods seem to be simple yet effective in balancing the trade-off between KL divergence and reward.
- Extensive numerical results are provided to justifiy the proposed method.

**Weaknesses:**

- The main drawback of this paper is that techniques like exponential moving average and SLERP are well-known, which renders the technical novelty insufficient.
- Despite extensive numerical evaluation, it is unclear why model averaging helps balance the trade-off between KL divergence and reward. According to the reviewer’s understanding, achieving this trade-off effectively requires the policy to explore intelligently and update gradually during initialization. However, it is unclear how this relates to the proposed technique or if there are other possible mechanisms and explanations.

**Questions:**

Some experimental details are unclear:

- In Table 1, the reviewer would like to know who is responsible for evaluating the side preference rate. Is it conducted by GPT-4 or by human labelers? Additionally, could you report the response length? If judged by GPT-4, the win rate may be more easily biased by the response length.

- The reviewer would like to know what kind of dataset is used for RLHF. It is quite surprising to see improved performance in the benchmarks shown in Table 2. The results in the table seem to suggest there is no alignment tax.
- The main numerical evaluation in this paper is the reward. Since only Table 2 appears to show the benefit of KL (I assume), could the authors provide evaluation results of benchmarks in Table 2 at different points along the KL-reward trade-off curve, or for configurations with different hyperparameters?

---

> ### Author Response · Authors · 2024-11-16
>
> We thank R.XQNe for their detailed review and for highlighting our results; we try to address the expressed concerns below.
>
> ## 1. Contributions (duplicate of R.ApSK.1)
>
> Rather than introducing a new weight averaging strategy, we (1) analyze the benefits of existing strategies for an important problem (KL-reward trade-off for RLHF of LLMs) and (2) propose a general framework, where three different weight averaging strategies are used at three different stages. We firmly believe our contributions are significant for the following reasons:
> - Using the **EMA as the anchor in the KL for RLHF of LLMs is novel**.
> - SLERP was indeed introduced in [Shoemaker1985], but as stated l.462, remains "relatively underexplored in the academic literature, with limited studies such as [Kim2024]". In Section 3.2 and Appendix B, we provide new **theoretical insights about the benefits of SLERP, explaining why it achieves higher rewards than LERP** (used in [Wortsman2022a]).
> - The idea of LITI is indeed inspired by [Wortsman2022b], from a different context (generalization in computer vision). Crucially, we introduce the novel idea of using the **LITI checkpoint for initializing subsequent RLHF iterations**, which significantly boosts performance.
>
> [Shoemaker1985] Animating rotation with quaternion curves.
>
> [Wortsman2022a] Model soups.
>
> [Kim2024] Token fusion: Bridging the gap between token pruning and token merging.
>
> [Wortsman2022b] Robust fine-tuning of zero-shot models
>
> ## 2. Why model averaging improves the trade-off between KL and reward (duplicate of R.pmz9.2)
>
> R.XQNe states that "this trade-off effectively requires the policy to explore intelligently and update gradually during initialization"; we fully agree on that, and actually each of our stage enables such **cautious exploration**.
>
> - Section 3.1: EMA "anchor induces a gradual automatic annealing and relaxation of the KL regularization. Specifically, the policy is initially strongly tied to the SFT initialization, and then progressively unleashed, allowing for more aggressive gradient updates later in training, leading to higher reward" (l.213).
> - Section 3.2: SLERP boosts rewards by merging the abilities of the different models.
> - Section 3.3: LITI "trades off between some newly acquired behaviors leading to high rewards vs. general knowledge from the SFT initialization", reduces forgetting and "recovers generalizable features from pre-training" (l.279).
> - Section 3.4: then applying those three stages iteratively enables to progressively move away from the SFT, with rewards getting higher at every iteration.
>
> All of these insights are ablated individually in respectively Section 4.1, 4.2, 4.3 and 4.4.
>
> Said differently,
> - EMA explores with automatic annealing,
> - SLERP merges the gained abilities,
> - LITI takes a step back,
> - and applying iteratively these 3 stages progressively explores higher-KL regions.
>
> As discussed in Section 6 from l.488 to l.500, this mimics an outer-inner optimization loop, where each RLHF finetuning ends-up being a small step in a more robust direction.
>
> ## 3. SxS setup
>
> The scores in Table 1 are computed with a **Gemini model as a judge, prompted to favor "short and concise generations"**. We also investigated a "length controlled SxS" score, similarly to [Dubois2024], where we regress the SxS score onto the response length delta; we obtain similar rankings among methods.
>
> Regarding R.XQNe 's request about "reporting the response length", they were already reported in Fig.18 from Appendix E. As a side note, we also explain in Appendix E how to potentially fix a potential length bias in WARP.
>
> [Dubois2024] Length-Controlled AlpacaEval: A Simple Way to Debias Automatic Evaluators

---

> ### Author Response · Authors · 2024-11-16
>
> ## 4. RLHF dataset
>
> The dataset used for RLHF comprises **a diverse set of prompts** designed to cover a wide range of capabilities and safety considerations. This includes prompts from conversation/chatbots datasets, but also prompts related to reasoning, mathematics and coding. The dataset is carefully curated to ensure diversity in topics, difficulty levels, and writing styles. The inclusion of reasoning prompts enables improving scores on certain benchmarks such as GSM8K (see next topic).
>
> ## 5. SxS and benchmark scores at different KL
>
> As requested by R.XQNe, we have added in the revised version of the paper an Appendix G named "SxS and benchmark scores at different KL". There, we show how **the performances in terms of reward-KL are transposed in terms of SxS and accuracy on real-world benchmarks**. To this end, we evaluate different checkpoints obtained by LITI between the SFT and the SLERP at the end of the 3nd iter. The results in the novel Fig. 20 confirms the importance of having high reward and low KL, to mitigate reward hacking and the alignment tax. Specifically, in Fig.20.a, we plot the SxS against Mistral v1, Mixtral v2 and Mixtral as a function of their KL to the SFT. We see that the SxS get better until a plateau at KL 110; then, results degrade due to **reward hacking**.
> In the Fig.20.b, we plot accuracies on GSM8K, MMLU, and MBPP as a function of their KL to the SFT; we see different behaviors. First, GSM8K actually increases for larger KL (because of larger reward); we suspect this is because we have some related prompts in the dataset. In contrast, scores on MMLU decreases, showing a critical **alignment tax**. Finally, scores on MBPP are stable.
>
> As a side note,
> - Table 1 already hinted at the importance of the KL; indeed, SxS performances are better at the 3nd iteration than at the 4th (where reward and KL are higher), showcasing reward hacking.
> - Fig.19 in Section F also confirms another flaw from larger KL; showcasing reduced diversity as the policy moves away from its initialization.

---

> > ### Comment · Reviewer_XQNe · 2024-11-26
> > **Thanks for your response**
> >
> > Thanks for your clarification for the method and experiment setting. I decide to maintain my rating at this stage.

---

> ### Author Response · Authors · 2024-11-26
>
> We thank R.XQNe for acknowledging our rebuttal. Essentially, the initial review suggested:
> - limited novelty; we clarified that using EMA as the anchor for RLHF is novel, that our insights on SLERP are novel, and that using the LITI checkpoint iteratively is novel.
> - more information about why WARP works; we linked information from the paper showing that EMA performs automatic annealing, SLERP boosts rewards and that LITI trades-off abilities.
> - more information about the evaluation; we provided them, while linking information that were already in the Appendix.
> - more information about the dataset; we provided them.
> - more SxS and benchmark scores at different KL; we provided them.
>
> If R.XQNe could please detail the possible remaining concerns, that would be helpful. Thank again for the time and consideration. Sincerely, authors.

---

### Official Review · Reviewer_ApSK · 2024-11-04

**Soundness:** 3
**Presentation:** 3
**Contribution:** 2
**Rating:** 5
**Confidence:** 4

**Summary:**

The paper improves the trade-off between KL and reward during RLHF by proposing Weight Averaged Rewarded Policies (WARP). WARP is designed to optimize the KL-reward Pareto front of solutions, by using three variants of WA at three different stages of the alignment procedure: Exponential Moving Average (EMA), Spherical Linear intERPolation of task vectors (SLERP) and Linear Interpolation Towards Initialization (LITI).

**Strengths:**

- The paper is clearly written and easy to understand, using exponential moving average of the base policy in RLHF should be less conservative than a fixing target.
- The empirical analysis is robust, featuring comparisons against state-of-the-art models and demonstrating WARP's effectiveness in balancing KL regularization and reward optimization.
- The paper considers a wide variety of tasks to demonstrate potential improvements.

**Weaknesses:**

- Application of existing techniques to RLHF: The novelty is limited, the techniques used in the paper are almost proposed from the deep learning or reinforcement learning literature. For example, EMA anchor used ideas from trust-region updated reinforcement learning algorithms like TRPO [1]; SLERP uses idea from model merging by weight averaging [2,3]. LITI uses idea from WiSE-FT [4]. Given previous work that already applies similar ideas to reward modeling RLHF [5], the novelty is further weakened. A rigorous theoretical framework to justify why applying these techniques to the policy is better than reward is missing.
- Ablation studies: The paper introduces three different procedures, it's better to provide a detailed ablation studies to each procedure to verify no one procedure is useless.
- Training WARP is costly, requiring multiple iterations and multiple RL runs at each iteration. Also, more hyperparameters are introduced, making the algorithm more complex.

[1] Trust region policy optimization.

[2] Model soups: averaging weights of multiple fine-tuned models improves accuracy without increasing inference time.

[3] Diverse weight averaging for out-of-distribution generalization.

[4] Robust fine-tuning of zero-shot models.

[5] WARM: On the Benefits of Weight Averaged Reward Models.

**Questions:**

1. Why should we have the Linear Interpolation Towards Initialization (LITI) step? This step takes an interpolation
from the merged model towards the initialization as the next iteration's initialization. Why do we need that? Can't that be implemented by adjusting the \beta in KL regularization with updated base policy and using only one iteration?
2. In Algorithm 1, is line 13 and the ouput using the same \eta?

---

> ### Author Response · Authors · 2024-11-16
>
> We would like to thank R.oGVx for their review and questions that we try to answer below.
>
> ## 1. Contributions (duplicate of R.XQNe.1)
>
> Rather than introducing a new weight averaging strategy, we (1) analyze the benefits of existing strategies for an important problem (KL-reward trade-off for RLHF of LLMs) and (2) propose a general framework, where three different weight averaging strategies are used at three different stages. We firmly believe our contributions are significant for the following reasons:
> - Using the **EMA as the anchor in the KL for RLHF of LLMs is novel**. As a side note, we already acknowledged l.202 that "in RL (notably for control tasks) it is common to regularly update the anchor" (though the TRPO paper did not use EMA).
> - SLERP was indeed introduced in [Shoemaker1985], but as stated l.462, remains "relatively underexplored in the academic literature, with limited studies such as [Kim2024]". In Section 3.2 and Appendix B, we provide **new theoretical insights about the benefits of SLERP, explaining why it achieves higher rewards than LERP** (used in [Wortsman2022a]).
> - The idea of LITI is indeed inspired by [Wortsman2022b], from a different context (generalization in computer vision). Crucially, we introduce the novel idea of using the **LITI checkpoint for initializing subsequent RLHF iterations**, which significantly boosts performance.
>
> [Shoemaker1985] Animating rotation with quaternion curves.
>
> [Wortsman2022a] Model soups.
>
> [Kim2024] Token fusion: Bridging the gap between token pruning and token merging.
>
> [Wortsman2022b] Robust fine-tuning of zero-shot models.
>
> Regarding weight averaging for reward models [Rame2024], as stated l.476, "WARP is conceived as a response to WARM, demonstrating that model merging can tackle two key RLHF challenges; policy learning in WARP and reward design in WARM". Specifically, [Rame2024] improved reward modeling by leveraging the benefits of WA in terms of reduction of variance and memorization. In contrast, we improve policy learning by leveraging other benefits of WA: as a dynamic anchor in EMA, to combine abilities with SLERP and to reduce forgetting in LITI. This means that the **empirical contributions are complementary**, and WARM and WARP should actually be used together. From a **theoretical perspective, our insights from Appendix B are novel**, showcasing the benefits of WA to handle a problem specific to policy learning for a given reward: the KL-reward trade-off.
>
> [Rame2024] WARM: On the benefits of weight averaged reward models
>
> ## 2. Ablation studies
>
> We respectfully disagree with R.oGVx's assessment: **our ablation studies in Section 4 already demonstrate that each stage, individually, improves performances**. Specifically:
> - Section 4.1 for EMA: notably Fig 3.b shows that using an EMA anchor improves the KL-reward trade-off for a single RLHF run (so without SLERP or LITI).
> - Section 4.2 for SLERP: notably Fig.3.c shows that merging $M=2$ models with SLERP increases the reward (even without LITI), and in particular more than with LERP.
> - Section 4.3 for LITI: notably Fig.4.b shows that even with $M=1$ model (so without SLERP), LITI still performs better than the checkpoints along RLHF.
>
> You can also find other ablations in Appendix:
> - Appendix D.1/D.2 and Fig.12/Fig.14 for EMA.
> - Appendix C and Fig.8/Fig.9 for SLERP.
> - Appendix D.3/D.4 and Fig.10/Fig.13/Fig.16/Fig.17 for LITI.
>
> This is acknowledged by R.pmz9 who states that "the ablation study for each stage is given".
>
> ## 3. Increased training cost (duplicate of R.HK4g.1)
>
> We agree that WARP is indeed more costly than traditional single-run strategies. This limitation was discussed from l.513 to l.520 in the Section 6 of our submission: "we see WARP as a fine-tuning method that can transform additional compute allocated to alignment into enhanced capabilities and safety. This would allow scaling (the traditionally cheap) post-training alignment, in the same way pre-training has been scaled. Indeed, historically, pre-training efforts have benefited much more from compute scaling, fine-tuning efforts remaining significantly cheaper."
>
> For example, the pre-training of the Gemma 7B required 6T tokens, several orders of magnitude above a traditional RLHF run, which would anyway fail to leverage additional compute; if we simply add more training steps, then the KL increases and polices hack the reward and/or suffer from an alignment tax. **WARP is a novel strategy that could enable scaling the post-training phase**. And finally, as stated in conclusion l.535, "we hope WARP could contribute to safe and powerful AI systems by scaling alignment".
>
> Regarding the additional hyperparameters, there are two of them, the EMA update rate $\beta$ and the LITI update rate $\eta$, and we show results are robust to their values in respectively Fig.15 and Fig.16.

---

> ### Author Response · Authors · 2024-11-16
>
> ## 4. Importance of LITI (duplicate of R.HK4g.2)
>
> LITI is empirically useful because, as described in Observation 1, it "reveals a better Pareto front than the one revealed during RL fine-tuning" (l.274). We consistently observe this, no matter the number of training steps in Fig.4.a, the number of merge models in Fig 4.b or the iteration number in Fig.4.c. Should we find a better $\beta$ for the KL, then we believe that the LITI would perform even better.
>
> To understand this empirical success, we highlighted 3 benefits of LITI in Section 3.3 based on previous works:
> 1. following [Wortsman2022b], LITI "recovers generalizable features from pre-training that might be lost during fine-tuning" (l.278).
> 2. following [Lin2024], LITI "increases feature diversity, efficiently balancing between generality and task specificity" (l.279).
> 3. following [Jang2024], LITI approximates the "geometric projection of the ideal weights [which] is located between the merged model and the initialization" (l.281).
>
> Overall, the literature suggests that **LITI efficiently trades-off between the initial abilities of the initialization, and those acquired during fine-tuning**.
>
> We provided novel theoretical insights about it in Lemma 5 from Appendix B.2.3, based on the concavity of the reward and convexity of the KL. As highlighted in Fig.7, we show the "LITI policy has a higher reward than the interpolated reward at a lower KL." (l.1210).
>
> Yet, full theoretical comparison against the checkpoints collected along RLHF would require new theoretical tools. Intuitively, from an optimization perspective, fine-tuning is an unstable procedure with variance and noise; in contrast, the task vector (the difference between the final checkpoint and the init) provides a more consistent direction as it is averaged over multiple batches sampled sequentially. Then, whereas the checkpoints along RLHF suffer from suboptimal trajectory, the LITI depends only on the latest checkpoint, with reduced variance. These findings are consistent with the generalization literature in computer vision, notably Fig.3 from [Wortsman2022b], where LITI is systematically above the optimization trajectory in terms of in-distribution/out-of-distribution trade-off.
>
> [Wortsman2022b] Robust Fine-Tuning of Zero-Shot Models.
>
> [Lin2024] Mitigating the Alignment Tax of RLHF.
>
> [Jang2024] Model Stock: All We Need Is Just a Few Fine-Tuned Models.
>
>
> ## 5. Notation in Algorithm 1
>
> We thank the reviewer for highlighting this lack of clarity; in Algorithm 1, the $\eta$ in line 13 and output do not share the same value, as the one in the output varies between $0$ and $1$ to reveal the final Pareto front. The **notation is changed** in the revised version of the paper.

---

### Author Response · Authors · 2024-11-16
**General response to reviewers**

We would like to thank the reviewers for their feedback. We are glad they found our paper "clearly written and easy to understand" (R.ApSK), that "the intuition [...] and the ablation study for each stage is given" (R.pmz9), leading to "robust" (R.ApSK) and "extensive numerical results" (R.XQNe), "outperforming open-source models" (R.HK4g).

We also received the expressed concerns, and tried to address them in response to each reviewer individually. Moreover, we made some minor modifications to the manuscript, notably:
- moving a paragraph named "Weight averaging" from Section 5 to Section 2, to provide earlier "some more introduction for weight averaging", as requested by R.pmz9.
- adding an Appendix G named "SxS and benchmark scores at different KL", as requested by R.XQNe, where we show how the performance in terms of KL-reward is transposed in terms of SxS and accuracy on real-world benchmarks.

Should there be any remaining questions, we are happy to provide further clarification.
Sincerely, authors.

---

> ### Author Response · Authors · 2024-11-26
> **End of discussion period**
>
> Dear Reviewers,
>
> This is a friendly reminder about our paper. We understand you may have many papers to review, but we would appreciate you taking the time to consider our rebuttal. We believe we addressed all questions and concerns expressed in the initial reviews; if any point requires further clarification, we remain happy to engage before the end of the discussion period.
>
> Thank you once again for your time and consideration.

---

> ### Author Response · Authors · 2024-12-03
>
> Dear Reviewers and Area Chair,
>
> As the discussion period ends today, we take the opportunity to summarize our perspective on the main concerns from the reviewers, and our provided responses.
>
> - Limited novelty (R.ApSK.1 and R.XQNe.1). We have three main contributions: using EMA as the anchor for RLHF, new insights on SLERP, and using the LITI checkpoint as initialization iteratively.
>
> - Ablations (R.ApSK.2) and why WARP improves the KL-reward trade-off (R.XQNe.2 and R.pmz9.2). We answered with quotes and experiments from the paper, and showed that EMA performs automatic annealing, SLERP boosts rewards and LITI trades-off between initial abilities and those acquired during fine-tuning (R.ApSK.4 and R.HK4g.2).
>
> - Increased computational cost (R.ApSK.3 and R.HK4g.1). Indeed WARP requires more training compute for better performance. We see WARP as a strategy that could scale the post-training phase (similarly to how the pre-training was scaled), transforming additional compute dedicated to alignment into enhanced capabilities and safety.
>
> - Additional information. We revised our paper to provide more information weight averaging (R.pmz9.1), detailed our experimental setups (R.XQNe.3, R.XQNe.4), and provided more results (R.XQNe.5) validating our insights.
>
> Then R.HK4g stated that "The authors' response effectively addressed my question".
>
> Based on this, we hope we answered the questions and tackled the expressed concerns.
>
> Thank you for your time and consideration. Sincerely, Authors.

---

### Meta-Review · Area_Chair_1Sc6 · 2024-12-19

**Metareview:**

The paper proposes a new RLHF algorithm that merge policies in the weight space to address the trade-off between KL and reward. This is a borderline paper without a strong champion. The disagreements  from the reviewers are

1. Novelty: Many techniques used here have already been proposed in RL, but the authors argue that they have not been used in the RLHF context with LLM.

2. Complexity of the proposed method. The proposed algorithm has 3 steps. Reviewers have raises issues on the missing theoretical rigorousness on these steps are designed and support on why they indeed work as the authors claim intuitively. There're also debates on whether the ablation experiments are sufficient to make up for the lack of theory.

3. Increased computed cost. The proposed require training multiple policies, which cost more than some other RLHF finetuning scheme. Nonetheless, the authors suggest this is a feature of scaling post-training.

For point 1, I think identifying the effectiveness in the LLM context warrants its own novelty. For point 2, however, I agree with the reviewers that not enough evidence is provided to support the intuition claims. There're not full theory that directly analyze WARP's performance but just lemmas for different parts at a high level. More explanation is needed also to explain why these 3 steps are designed to combine in the current way. To fully  support these claim empirically without theory, I think experimental results on multiple LLMs, datasets and labeling schemes are needed. For point 3, the authors should provide more experimental results to support the effectiveness of the scaling argument, to justify indeed the proposed way is a better way to scale the post-training compute.

Overall, I think there is some room for improvement from the current paper to warrant a clear acceptance.

**Additional Comments On Reviewer Discussion:**

Reviewer ApSK raises concerns on limited novelty, limited theoretical analysis, detailed ablation, and the introduction of more HPs. After the discussion, the concerns on missing of novelty and principled reasoning of why the framework is designed remains.

Reviewer XQNe raises concerns on novelty, and the lack of explanation of the proposed method's working mechanism. Despite clarification on the method and the experiments, the rating remains.

Reviewer pmz9 raises concerns on missing of working mechanism. The authors expand with more insights

Reviewer HK4g raises concerns on increased computational cost and missing of working mechanism. The responses address the concerns.

---

### Decision · Program_Chairs · 2025-01-22

Reject